# Experimental investigation of the interplay between transverse mixing and pH reaction in porous media

Adi Biran[1], Tomer Sapar[1], Ludmila Abezgauz[1], and Yaniv Edery[1]

[1]Technion – Israel Institute of Technology, Haifa, Israel

**Correspondence:** Yaniv Edery (yanivedery@campus.technion.ac.il)

**Abstract.** pH-induced reactive transport in porous environments is a critical factor in Earth sciences, influencing a range of natural and anthropogenic processes such as mineral dissolution/precipitation, adsorption/desorption, microbial reactions, and redox transformations. These processes, pivotal from carbon capture and storage (CCS) applications to groundwater remediation, are determined by pH transport. However, the uncertainty in these macroscopic processes' stems from pore-scale heterogeneities and the high diffusion value of the ions and protons forming the pH range. While practical for field-scale applications, traditional macroscopic models often fail to accurately predict experimental and field results in reactive systems due to their inability to capture the details of pore-scale pH range. This study investigates the interplay between transverse mixing and pH-driven reactions in porous media. It focuses on how porous structure and flow rate affect mixing and chemical reaction dynamics. Utilizing confocal microscopy, the research visualizes fluorescently labeled fluids, revealing variations in mixing patterns from diffusive in homogenous to shear-driven in heterogeneous media. However, pH-driven reactions show a different pattern, with a faster reaction rate, suggesting quicker pH equilibration between co-flowing fluids than predicted by transverse dispersion or diffusion. The study highlights the unique characteristics of pH change in water, which significantly influences reactive transport in porous media.

## 1 Introduction

The distribution of most chemical species in a porous environment is generally determined by both transport and biogeochemical reactions, as described by the term reactive transport (Holzbecher (2005); Carrera et al. (2022)). Reactive transport is involved in diverse processes, either naturally or anthropogenically occurring, such as mineral dissolution and precipitation (Steefel and MacQuarrie (1996); Noiriel and Soulaine (2021); Stolze et al. (2022); Goldberg-Yehuda et al. (2022)), adsorption and desorption (Carrillo-González et al. (2006); Nützmann et al. (2005)), microbial reactions (Stocks-Fischer et al. (1999); Thullner et al. (2005)), and redox transformations (Thullner et al. (2005); Sposito (2008)). Reactive transport in porous media can be described with either pore-scale or Darcy-scale (macroscopic) models. Although pore-scale simulations have a solid physical foundation, they require knowledge of pore size distribution, geometry, tortuosity, and connectivity. These are seldom available and are impractical as predictive tools at scales that are orders of magnitude larger than the pore scale. Therefore, macroscopic models have developed to overcome these limitations (Battiato and Tartakovsky (2011); Valocchi et al. (2019); Ghaderi Zefreh et al. (2019)). Macroscopic representation is based on upscaling the porous medium by averaging it over space

and time scales in a representative elementary volume (REV), which allows the replacement of a solid-liquid domain with an equivalent continuum (Chiogna and Bellin (2013)). For the reaction to occur, both reactants must be in the vicinity of each other, and the process enabling them to ultimately react is the mixing, which is scale dependent (Alhashmi et al. (2015); Acharya et al. (2007); Guadagnini et al. (2009); Dentz et al. (2011); Al-Khulaifi et al. (2017); Rücker). However, the pore-scale mixing impacts the larger-scale reactive transport behavior (Datta et al. (2013); Browne and Datta (2023)). Due to this mixing multiscale nature, there is still a lack of understanding of the integration between coupled transport and reactions at multiple scales of the porous medium, which poses a challenge in predicting mixing-driven reactions (Edery et al. (2015); Tartakovsky et al. (2009); Borgman et al. (2023)). Thus, it is necessary to measure both mixing and reaction at the pore scale, regarding pore properties. It is particularly essential to understand how mixing patterns at the pore scale affect pH-driven chemical reactions, as these reactions are ubiquitous in porous media, such as soils and aquifers (Lai et al. (2015)). Examples of such reactions are, dissolution and precipitation of soil carbonates and sulfates (Sposito (2008)), nitrification and denitrification processes (Ward et al. (2011); Edery et al. (2011, 2021); Shavelzon and Edery (2023)), protonation and deprotonation of carboxyl and phenolic groups in soil organic matter (Sparks et al. (2024)). Soil pH has an enormous influence on soil biogeochemical processes, as it influences the solubility of plant nutrients, phytotoxic elements, and pollutants, and determines their biological availability and mobility (Penn and Camberato (2019); Neina (2019); Dehkharghani et al. (2019)). Specifically for pH reactions, experimental data with high Peclet value for transverse reaction are in good agreement with the Advection-Dispersion-Reaction equation (ADRE), which uses a single diffusion coefficient for all species in a multispecies reactive system (Loyaux-Lawniczak et al. (2012)), especially in stirred flow-cell reactors (Liu et al. (2011)). Considering the coupling between mixing and reactive transport processes and how both are scaled with the heterogeneity, specifically in the context of pH reactions in heterogenous soil, a set of experiments is proposed to observe if, indeed, the same coupling between mixing and reaction occurs for pH spread and reactions. These experiments focus on investigating how porous medium layouts ranging from homogeneous to heterogeneous affect pH-driven reactions by examining the pattern of transverse dispersion of co-flowing fluids for both mixing and pH. This is done by tracking the mixing and pH spread for two Peclet values using fluorescently labeled fluids imaged by a confocal microscope. The mixing experiments showed that transverse mixing varies from diffusive mixing in the homogeneous case to shear-driven mixing in the heterogeneous case. However, the pH measured in the pH experiments does not follow the pH value calculated from the mixing pattern. Instead, it shows a larger spread, suggesting that the co-flowing fluids' pH difference equilibrates faster than the mixing. We identify the proton transfer mechanism, which is comparatively faster than the transverse dispersion or diffusion, as the dominant mechanism, especially for lower Peclet. Pore-scale simulations agreed well with the mixing experiments and provided reasonable results for the pH experiments after considering the enhanced diffusion due to the proton transfer mechanism.

## 2 Methods

To investigate how the porous structure and flow rate influence mixing and chemical reaction, three sets of experiments are employed to visualize mixing and reaction in a porous medium. The first set is mixing experiments, where a conservative tracer

is used to test the effect of different pore size variations (heterogeneities) with different flow rates on the local mixing dynamics.

In the second set a reactive experiment is employed under the same conditions as the conservative experiments, where the pH reactant is uniformly distributed at the flow cell, and only the pH is unevenly distributed. And in the third set, both the pH and pH reactant are unevenly distributed in the cell to examine the effect of mixing on neutralization reaction dynamics, resulting in pH change, under the same conditions.

## 2.1 Experimental setup

All sets of experiments, shown in Figures 1.a.-c., were performed in a Polydimethylsiloxane (PDMS) microfluidic flow cell, sized $\sim 4.5$ mm $\times$ 1.3 mm $\times$ 0.05 mm. Each cell was composed of $\sim 300$ cylindrical pillars ($R = 50 \mu$m) so the internal porosity of the cell was $60\% \sim 70\%$ (see Table 1). The tracer moved only in the pore space among the pillars, which were set in four different arrangements within the cell to achieve four different levels of heterogeneity: from completely homogeneous where the pillars center were set on a perfect lattice grid with a normalized standard deviation of $\sigma/R = 0$, to the most heterogeneous

arrangement where the pillars centers were randomly moved in the x and y direction following a Gaussian distribution with a normalized standard deviation of $\sigma/R = 0.5$ (see Figures 1.d.).

Each cell had two parallel inlets (right and left), each of them set at 425 $\mu$m from the edge of the cell, and one funnel shaped outlet. At the two outlets, a syringe pump (Chemyx Fusion 200 Two Channel model) with a small diameter glass syringe (100 $\mu$L Hamilton glass syringe) allowed a continuous movement for the motor and the piston with no oscillations for the applied

fluxes (100, and 10 $\mu$L/h flow rate, resulting in a Darcy velocity of $v_d = 0.142$, and 0.0142 $cm/s$, respectively). These two velocities provided two Peclet numbers (Pe), as depicted by the following equation:

$$Pe = \frac{v_d R}{D} \tag{1}$$

The Peclet number is a measure of the velocity magnitude ($v_d$), and the diffusion ($D$), which is an intrinsic property of the fluids over the mean pore size ($R$) (Bossis and Brady (1987)). While the mean pore size remains the same for all heterogeneity, there

are small porosity ($\phi$) variations (see Table 1 for details). However, the main heterogeneity effect is on the interface between the co-flowing fluids, forming a torturous path. To address this, we define an effective diffusion coefficient $D_{eff} = \frac{D\phi}{T}$, which scales the diffusion of the reactants in water, as shown in many studies (Ray et al. (2018); Fogler (2011); Guo et al. (2022); Kim et al. (1987); Quintard (1993); Quintard and Whitaker (1993); Beyhaghi and Pillai (2011)). The tortuosity can be directly calculated from the normalized standard deviation $\sigma/R$, which marks the range for the pillar center movement from a uniform

grid using the following relation, $T = 1 + \sigma/R$ (as shown in Eliyahu-Yakir et al. (2024)), and leading to the effective Pe number of:

$$Pe_{eff} = \frac{v_d R T}{D\phi} \tag{2}$$

and scaling the Peclet number as depicted in Table 1.

The fluorescent conservative tracer used for the mixing experiments (Figure 1.a.) is rhodamine 6G (R6G), which is widely

used to visualize flow patterns, such as in the domain of environmental hydraulics (Barzan and Hajiesmaeilbaigi (2018)).

| $\sigma/R$ [−] | 0.0 | 0.01 | 0.1 | 0.5 |
|---|---|---|---|---|
| $\phi$ [−] | 0.68 | 0.64 | 0.64 | 0.62 |
| $T$ [−] | 1 | 1.01 | 1.1 | 1.5 |
| $Pe_{eff}/Pe$ [−] | 1.47 | 1.58 | 1.72 | 2.42 |

**Table 1.** The table depicts the porosity, tortuosity, and effective Peclet ratio for each heterogeneity.

Pyranine (8-hydroxypyrene-1,3,6-trisulfonate) is used for the reactive and combined experiments (Figure 1.b.-c.) as the pH reactant, as its fluorescent emission spectra and intensity are highly dependent on medium pH (Avnir and Barenholz (2005)), therefore suitable for monitoring pH changes. Double distilled water (DDW) purified by Milli-Q, with $\approx 18$ M$\Omega$·cm$^{-1}$ at lab temperature of 25 °C, is the used in both the R6G and reactive experiments. The R6G's concentrations were 2 mg/50 mL DDW (corresponding to 0.083 mM) and 9 mg/50 mL DDW for the pyranine (corresponding to 0.347 mM). These concentrations had no measurable effect on the fluid viscosity and density in this experimental setup.

All the experiments' pH values used for the reactive experiments were 7.3 and 12.3, resulting in higher and lower emission intensities, respectively. As such, they are related to their respected tracer and background solution. To achieve the wanted pH, we added a strong acid or a strong base (hydrochloric acid and sodium hydroxide respectively), to the pyranine aqueous solutions. When HCl is added, it ionizes to form the hydronium ion:

$$\text{HCl}_{(aq)} + \text{H}_2\text{O} \leftrightarrow \text{H}_3\text{O}^+_{(aq)} + \text{Cl}^-_{(aq)} \tag{3}$$

When NaOH is added, it ionizes to form the hydroxide ion:

$$\text{NaOH}_{(aq)} \leftrightarrow \text{Na}^+_{(aq)} + \text{OH}^-_{(aq)} \tag{4}$$

When set together, the hydronium and hydroxide ions react to form water in a neutralization reaction:

$$\text{H}_3\text{O}^+_{(aq)} + \text{OH}^-_{(aq)} \leftrightarrow 2\text{H}_2\text{O} \tag{5}$$

While the pyranine ($\text{ROH}_{(aq)}$) reaction is mainly with the $\text{OH}^-_{(aq)}$ as the reactive experiment is performed under basic pH:

$$\text{ROH}_{(aq)} + \text{OH}^-_{(aq)} \leftrightarrow \text{RO}^-_{(aq)} + \text{H}_2\text{O} \tag{6}$$

and, therefore, the intensity is mainly decreasing with the pH change (Figure 1.e.).

To perform the mixing and reactive experiments, we saturate the flow cell with the background solution, i.e., DDW for the mixing experiments, and pyranine solution at a pH of 12.3 (which will be regarded as basified pyranine from here on) for the reactive experiments. Subsequently, a 100 $\mu$L glass syringe, filled with the R6G, or pyranine at pH of 7.3 (which will be regarded as acidified pyranine from here on), is connected directly to the left inlet to reduce the experimental time until the R6G/acidified pyranine reaches the cell and forms an interface with the DDW/basefied pyranine.

A picture of the cell, filled with the background solution, is taken before the insertion of the R6G/acidified pyranine, providing a base image for the image analysis calibration. Thereafter, both 100 $\mu$L syringes with the R6G/acidified pyranine and

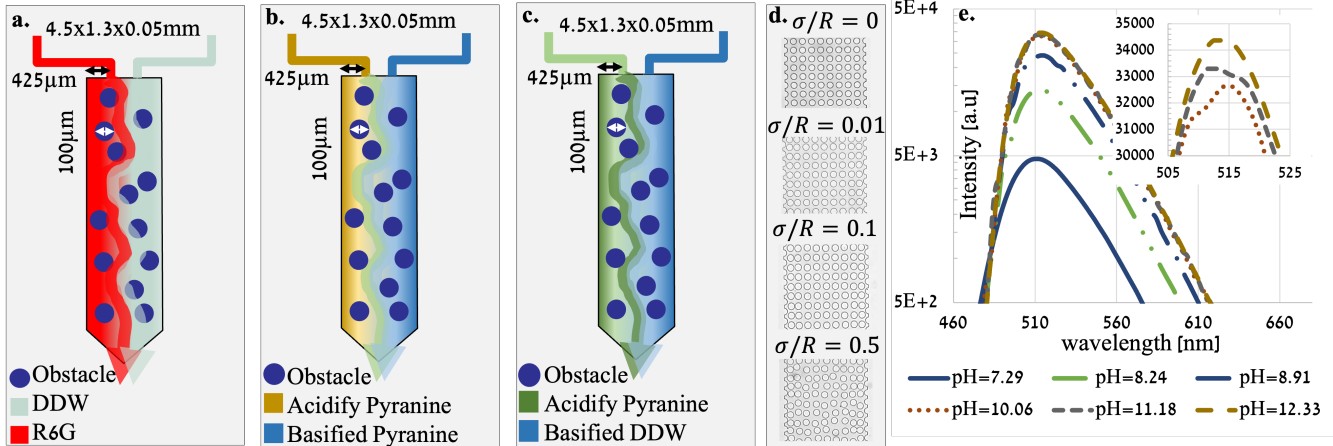

**Figure 1.** a. An illustration of the mixing experiment setup. b. An illustration of the reactive experiment setup (pH gradient only). c. An illustration of the combined experiment setup (pyranine concentration gradient and pH gradient). d. Four different pore size variations (heterogeneities) of the flow cell, from the homogeneous one ($\sigma/R = 0$) to the most heterogeneous ($\sigma/R = 0.5$). e. Intensity of pyranine emission on a logarithmic scale versus wavelength for various pH, as measured by UV-vis, and verified under the confocal. The inset is a blow-up on a linear scale to present the relevant separation of the pH to intensity.

DDW/basefied pyranine are placed in the same syringe pump with a pre-defined flow rate. This way, the DDW/basefied pyranine flowed from the right inlet, and the R6G/acidified pyranine from the left inlet had the same flow rate while interfacing roughly in the middle of the cell.

Changes in color intensity of the R6G occurred due to mixing (or dilution) with the DDW, while for the acidified pyranine, the intensity change is due to a pH change followed by the neutralization reaction. Finally, the cell is saturated manually with the R6G/acidified pyranine for mixing/reactive experiments from both inlets to produce a high-intensity, final image with known concentration for the image analysis calibration. The third set of experiments combines both concentration gradient and pH gradient, using DDW at a pH of 12.3 as the background solution (which will be regarded as basefied DDW from here on) and acidified pyranine as the reactive tracer. pH values were achieved using NaOH and HCl, similar to the reactive experiments. The combined experiments are made in the same process mentioned above, but only within the completely homogeneous medium and the most heterogeneous one, to present the effect of simultaneous migration of pyranine and pH.

## 2.2 Imaging setup

For both the R6G and reactive experiments, a confocal microscope (Nikon Eclipse Ti2-FP) was used to visualize the intensity change due to the mixing and reaction within the flow cell. The R6G is excited by a 546 nm laser, and tracked with the emission wavelength of 600 nm, while the pyranine is excited by a 405 nm laser and tracked with the emission wavelength of 550 nm.

All experimental images taken by the confocal are taken by a Prime BSI camera with a 95% quantum efficiency and $1e^-$ median noise, with an exposure time of 500 ms, bit depth of 16-bit, and magnification of x2.

For the 100 $\mu$L/h flow rate, a series of 50 pictures were taken 5 minutes after forming a stable interface between the fluids. Then, after an additional 5 minutes of delay, another series of 50 pictures is taken, under the same conditions. The two series of images are compared to verify the stability of the interface. For the 10 $\mu$L/h flow rate, the same imaging sequence was performed, with an initial time of 10 minutes and a subsequent delay time of 10 minutes. For both flow rates, each pixel intensity (marked as $I_{ij}$, for location $ij$) at each 50 pictures sequence, the variance of intensity per pixel did not exceed the 0.1% white noise of the camera. To verify that the interface among image sequences is stable, the criteria was set that the difference between the initial and later imaging sequence that exceeded the 0.1% (white noise of the camera) was averaged in absolute terms, and the stability of the interface was established if the average difference was isotropic and smaller than 1% (namely, $\left\langle \frac{|I_{ij}(t=5)-I_{ij}(t=10)|}{|I_{ij}(t=10)|} > 0.1\% \right\rangle < 1\%$), a similar analysis was performed around the interface to verify that the 1% difference is not the outcome of the bulk behavior. A MATLAB image processing program is developed to convert the image intensity received in the mixing experiments to normalized R6G concentration. Similarly, a program is developed to convert the image intensity received in the reactive experiments to its pH values. As such, this intensity analysis, which provides both the error bounds and repeatability of the layout, is done for both the R6G and pH experiment.

### 2.2.1 Imaging the mixing experiments

Conversion of image intensity to normalized R6G concentration is based on the Beer-Lambert law, dictating a linear relationship between the concentration and the absorbance of the solution (Barzan and Hajiesmaeilbaigi (2018)). The maximum and minimum intensity images are set to establish the scale between the maximum and minimum R6G concentration. The difference between each intermediate intensity and the minimal intensity is normalized to the difference between maximum and minimum intensities, yielding a unitless number between zero to one, i.e., the normalized R6G concentration:

$$C_{ij} = \frac{I_{ij} - I_{ij}(\text{min})}{I_{ij}(\text{max}) - I_{ij}(\text{min})} \ [-] \tag{7}$$

Recall that $I_{ij}$ is the image intensity at pixel $ij$, $I_{ij}(\text{min})$ is the intensity of the background solution image (DDW with no R6G), and $I_{ij}(\text{max})$ is the intensity of the R6G itself (DDW saturated with R6G) image. The validity of the method was verified for our setup as well as in other studies (Eliyahu-Yakir et al. (2023); Barzan and Hajiesmaeilbaigi (2018)).

The change in local normalized concentration ($C_{ij}$) for the R6G and DDW mixing can be transformed to pH and compared to the acidified and basified pyranine mixing. As the pyranine emission amplitude changes with the OH$^-$ groups, we base the calculation of pH on the OH$^-$ migration. This is done by the equation below:

$$\text{pH}_{\text{calculated}} = -(14 - \log[C_{ij} \cdot 10^{-(14-\text{low pH})} + (1 - C_{ij}) \cdot 10^{-(14-\text{high pH})}]) \tag{8}$$

where low pH and high pH are the pH values of the acidified and basified pyranine solutions.

### 2.2.2 Imaging the Reactive experiments

Unlike the R6G experiment, the intensity change of the pyranine due to the pH does not scale linearly; thus, a scheme of the process of converting raw data to pH distribution is developed for this study (shown in Figure 2). For the conversion of image intensity to pH, it was necessary to find a correlation between the two. To create a calibration curve, samples of pyranine dissolved in DDW (0.347 mM as in the experiments) at different pH values were made using HCl and NaOH. The flow cell was manually saturated with a sample with known pH, and an image of the cell was taken. The mean intensity of each image was then calculated.

The correlation between pH and mean image intensity was fitted (MATLAB Curve Fitting Tool application), in which a descending exponential function was set to fit the received calibration curve ($R^2 = 0.976$), shown in Figure 2.b. The equation is as follows:

$$y = -e^{a \cdot x} + b \tag{9}$$

where $y$ corresponds to the image mean intensity, $x$ corresponds to the pH value, while $a$ establishes the decedent rate ($a = 0.4977$), and $b$ is the maximum intensity ($b = 935.7$), and both are fitting parameters. This rapid exponential change in intensity due to pH marks the sensitivity of the pyranine to a narrow range of pH, which is reflected in the sharp transition between the pH values in Figure 2.c., and the following experimental pH results. The consumption of $OH^-$ by the pyranine will be negligible in changing the overall pH as the pyranine concentration is in equilibrium with the ions.

For the image analysis, we first fit a specific value of each of the parameters $a$ and $b$ in (9) to the intensity of each pixel composing the image. This is done by the two images produced at the beginning and the termination of each experiment, by cell saturation with acidified/basefied pyranine solution. Using image intensities of these two known pH values and (9), we find $a$ and $b$ matrices for each separate experiment and for each pixel, which is subsequently used for the conversion of image intensity to pH.

### 2.3 Comsol simulations

The results for both the mixing and reactive experiments, described in section 2.2, were simulated using the Comsol multiphysics Stokes flow simulator. To that end, the Autocad file with the 2D design and dimensions of the flow cells was imported to the simulator with their dimensions and no slip and no flow boundary condition for the pillars and walls. The inlet and outlet were defined as a Dirichlet boundary condition, corresponding to the constant flux condition imposed by the syringe pump. The simulation followed the following laminar flow equations for an incompressible fluid, namely the continuity, mass conservation, and viscous stress, respectively:

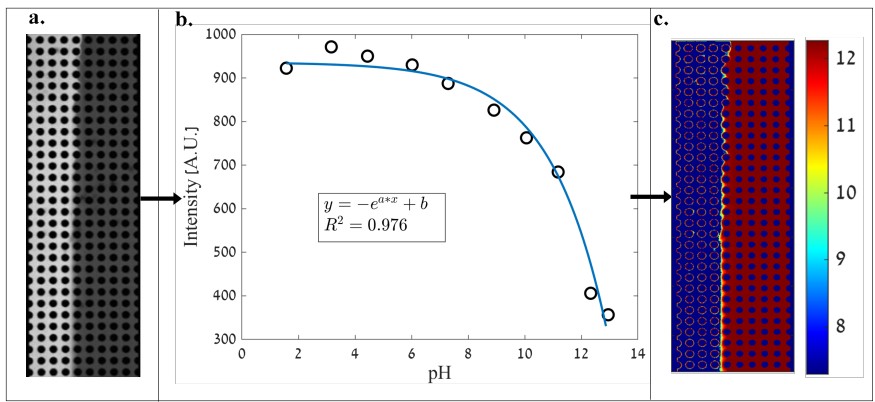

**Figure 2.** Scheme of the process of converting image intensity to pH distribution. a. Raw image, showing the intensity of acidified pyranine (0.347 mM) at pH of 7.3 on the left and basefied pyranine at pH of 12.3 on the right. b. Calibration curve showing the mean intensity of pyranine (0.347 mM) at different pH values, excited at 405 nm, as measured within the flow cell. c. Analyzed data image, showing pH distribution.

$$\rho\frac{\partial \mathbf{u}}{\partial t} + \rho(\mathbf{u}\cdot\nabla)\mathbf{u} = \nabla\cdot[-p\mathbf{I}+\mathbf{K}] \tag{10a}$$

$$\rho\cdot\mathbf{u} = 0 \tag{10b}$$

$$\mathbf{K} = \mu(\nabla\mathbf{u}+(\nabla\mathbf{u})^T) \tag{10c}$$

Where $\rho$ is the fluid density, $\mathbf{u}$ is the velocity in vector form (marked by bold) aligned and transverse $(^T)$ to the principal flow direction, $\nabla p$ is the pressure drop over the determinant $\mathbf{I}$, $\mathbf{K}$ is the stress tensor, and $\mu$ is the fluid viscosity. To account for the transport of the R6G and basefied solution, the following transport equation is used to account for the concentration $(C_n)$ of specific chemical species noted by $n$:

$$\frac{\partial c_n}{\partial t} + \nabla\cdot\mathbf{J_n} + \mathbf{u}\cdot\nabla c_n = R_n \tag{11a}$$

$$\mathbf{J_n} = -D_n\nabla c_n \tag{11b}$$

Where $\mathbf{J_n}$ is the diffusive flux calculated for each chemical species by its corresponding diffusion coefficient, $D_n$, and the chemical retardation factor per species $R_n$. The concentration, $C_n$, is inserted as $mol/M^3$ at the inlets according to the experimental values and as a fixed boundary value.

The maximum and minimum element sizes within the adaptive mesh used for the solid boundaries in the simulation are 1070 and 49.3 $\mu M$, while the maximum and minimum element sizes for the fluid calculation are 101 and 4.5 $\mu M$, for the adaptive mesh in the finite element linearized calculation. The simulation begins with the introduction of either the R6G or pH difference at the two inlets simultaneously, and allowing the simulation to evolve up to the initial time frame in the experiment

stated in section 2.1, namely 5 and 10 minutes for the Darcy velocity of $1.42$, and $0.142\ mm/s$, respectively, while the time discretization ranging between 5 to 15 seconds depending on the level of heterogeneity. The study state flow is achieved extremely fast within the simulation ($1 \sim 2$ simulated minutes), and therefore, there was no need to run it for another 5 and 10 minutes as in the experiment. These mesh sizes and temporal discretization were optimized to get the best results under the best stability, and simulations took about 5 minutes on a Core i5, 10 Gen computer with 16 Gig of memory. For each iteration, the concentration, velocity, and pressure were extracted, while the simulated 2D spatial distribution for the R6G and pH were compared with the experimental values using the 2D $R^2$ function in Matlab.

## 3    Results and Discussions

The results start in presenting the R6G experiments, showing how the heterogeneity level leads to various transverse mixing (3.1), followed by the measured pH experiments (3.2) and compared to the predicted pH, as calculated by the measured mixing. The experimental part is concluded by the presentation of the combined experiments (3.3). Finally, the results of the COMSOL simulations performed for the R6G and reactive experiments are presented (3.4).

### 3.1    Mixing experiments

As the R6G is inserted into the left side of the flow cell with a given flow rate, while the right side experiences the same flow rate with only DDW, we observe R6G migration between the sides due to the concentration gradient, via diffusion and transverse dispersion. The maximum normalized R6G concentration ($C_{ij} = 1$) is indeed at the left side, while the right side is at its minimum ($C_{ij} = 0$), as shown in Figure 3. Yet, the transition between the concentrations, representing the mixing due to diffusion and dispersion, varies according to the heterogeneity of the medium.

Both homogeneous and heterogeneous media show a relatively sharp interface between the R6G and the DDW near the flow cell inlet; this interface gradually disperses down-flow as the diffusion and dispersion propagate and drive the mixing between the fluids. However, this mixing mechanism, captured by the interface dispersion, varies in size and character from the homogeneous medium (Figures 3.a. and e.) to the most heterogeneous one (Figures 3.1.d. and h.). While in the homogeneous media, mixing is symmetrical within the cell, in the heterogeneous medium, mixing is determined by the pillar's setting and moves between different pores.

This change in mixing pattern demonstrates the different mechanisms governing the mixing as affected by the variations in pore size: mixing in the homogeneous medium is controlled mainly by diffusion, as shear forces effects are negligible, while in the heterogeneous medium, where pore size varies forming tortoise route among pillars, mixing is dominated by shear forces acting on the fluid close to the obstacle's boundaries. These forces result from the velocity gradient created due to the different pore sizes, where the smaller pores result in lower velocities and higher shear forces.

The Pe numbers calculated by (2) and presented in Table 1 are low yet still indicate that the velocity magnitude, which approximates the shear forces, has dominance on the diffusion in the pore scale, known to be critical in reactive transport (Nissan and Berkowitz (2019)). Mixing experiments results of the 10 $\mu$L/h flow rate (Figures 3.e.-h.) show that in all medium

heterogeneities, the interface between R6G and DDW is wider compared to the 100 $\mu$L/h flow rate, demonstrating the increased dominance of diffusion as the flow rate descends. Comparing heterogeneities of $\sigma/R = 0$, $\sigma/R = 0.01$, and $\sigma/R = 0.1$ (Figures 3.e.-g.), we see the more significant effect of diffusion in the homogeneous and nearly homogeneous medium, as mixing is apparent closer to the inlets. The most heterogeneous medium (Figures 3.h.) also shows a more dispersive pattern that encompasses several pore lengths and points to the more dominant role of shear over diffusion.

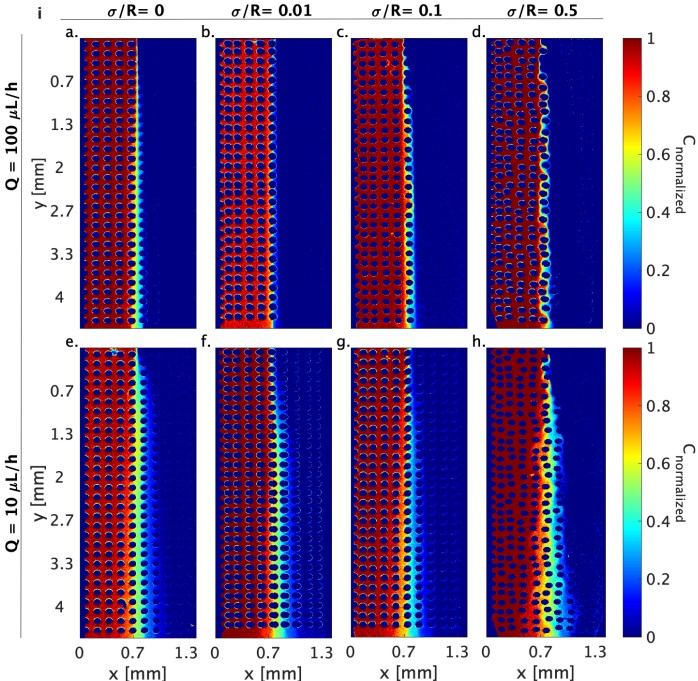

**Figure 3.** Mixing experiments depicting the distribution of the R6G normalized concentration for: a.-d. flow rate of 100 $\mu$L/h, and e.-h. 10 $\mu$L/h, for various medium heterogeneities.

### 3.2 Reactive experiments

Similarly to the mixing experiments, and following the experimental procedure described in 2.2.2 we performed reactive experiments with flow rates of 100 $\mu$L/h (Figures 4a.a.-d.) and 10 $\mu$L/h (Figures 4a.e.-h.). The mixing experiments provide the distinction between the role of diffusion and shear forces, where the first is manifested in the homogeneous, low-velocity case, and the latter is apparent in the heterogeneous and high-velocity case. However, the reactive experiments show that the pH reaction pattern does not necessarily follow the pH calculated from the normalized R6G concentration, particularly as the medium becomes more heterogeneous and the flow rate descends.

Of all the tested conditions, the patterns received in the homogeneous and the slightly heterogeneous ($\sigma/R = 0$ and $0.01$, respectively) media at a flow rate of 100 $\mu$L/h in the reactive experiments (Figures 4a.a.-b.) and the pH predicted by the mixing (Figures 4b.a.-b.), are relatively similar. In these conditions, the interface is almost symmetrical within the cell, although we do see a sharper gradient when the reaction occurs and a narrower interface. This sharp interface in pH value is probably due to the pyranine intensity exponential decay (Figure 2), and the logarithmic scale at which concentration is transforming to pH in (8).

Examining the reaction pattern at a flow rate of 100 $\mu$L/h in the more heterogeneous media, i.e., $\sigma/R = 0.1$ and $0.5$ (Figures 4a.c.-d.), clearly shows that the interface between the two fluids is not symmetrically distributed around the middle of the cell. Rather, it tends to migrate leftwards as the flow proceeds, indicating that the reaction occurs earlier and closer to the area of the acidified pyranine (7.3). This interface migration due to the diffusive nature of the $OH^-$ ions from high to low concentration occurs for all medium heterogeneities and was reported in previous studies on precipitation of $CaCO_3$ (Katz et al. (2011); Tartakovsky et al. (2007, 2008)), yet it becomes more dominant as the medium is more heterogeneous.

At a lower flow rate, sharper interfaces are received in the reactive experiments than those predicted by the mixing, at all medium heterogeneities (Figures 4a.e.-h. and 4b.e.-h., respectively). As seen in the 100 $\mu$L/h flow rate, this tendency is more noticeable as the medium becomes more heterogeneous. However, as heterogeneity increases, it appears that reaction tends to occur closer to the inlet at the lower flow rate. This is demonstrated in Figures 4a.c.-d. compared with Figures 4a.g.-h., where the latter presents a stronger migration of the interface so that a larger volume of the cell is occupied by the basified pyranine. This substantial migration of pH towards the acidified solution (recall that the pyranine concentration is uniform throughout the cell, and only the pH differs) cannot be the outcome of the initialization of solutions in the flow cell, as these measurements were taken after 100 and 20 pore-volumes for the 100 and 10 $\mu$L/h flow rates, respectively.

The calculated pH at 10 $\mu$L/h flow rate (Figures 4b.e.-h.) predicts a somewhat asymmetrical pattern regarding the basified vs. acidified pyranine distribution, and a slightly narrowing strip of the basified pyranine as fluids move towards the outlet zone, indicating that the reaction theoretically gets larger due to R6G diffusion. This increase is the outcome of the logarithmic scale of pH (see (8)), where the molar value of the access $OH^-$ ion are orders of magnitude higher on one side, which dominates over the cross-section. However, reactive experiments show that basified pyranine moves vertically along the cell significantly more than the calculated pH predicts, and the volume of the basified pyranine is increased at the expense of the acidified pyranine. This demonstrates that the reaction does not necessarily follow the mixing pattern in porous media, as the pH spreads faster than the R6G concentration predicts, an aspect that has a clearer representation in the following section.

### 3.2.1 Comparing the average pH transverse migration

Most studies and experiments do not have access to the pH or R6G spatial distribution, and therefore they rely on measuring the average values of pH in the system at a given volume. To reproduce this measurement, we further compare the averaged pH spread received in the reactive experiments with the averaged theoretical one we calculated according to the R6G concentration. Using a MATLAB program, we divided each analyzed image (of Figures 4b and 4a) into three, size-equal sections: inlet area, middle area, and outlet area (see Figure 4c 2.). In each section, we calculated the average pH of each column of the matrix

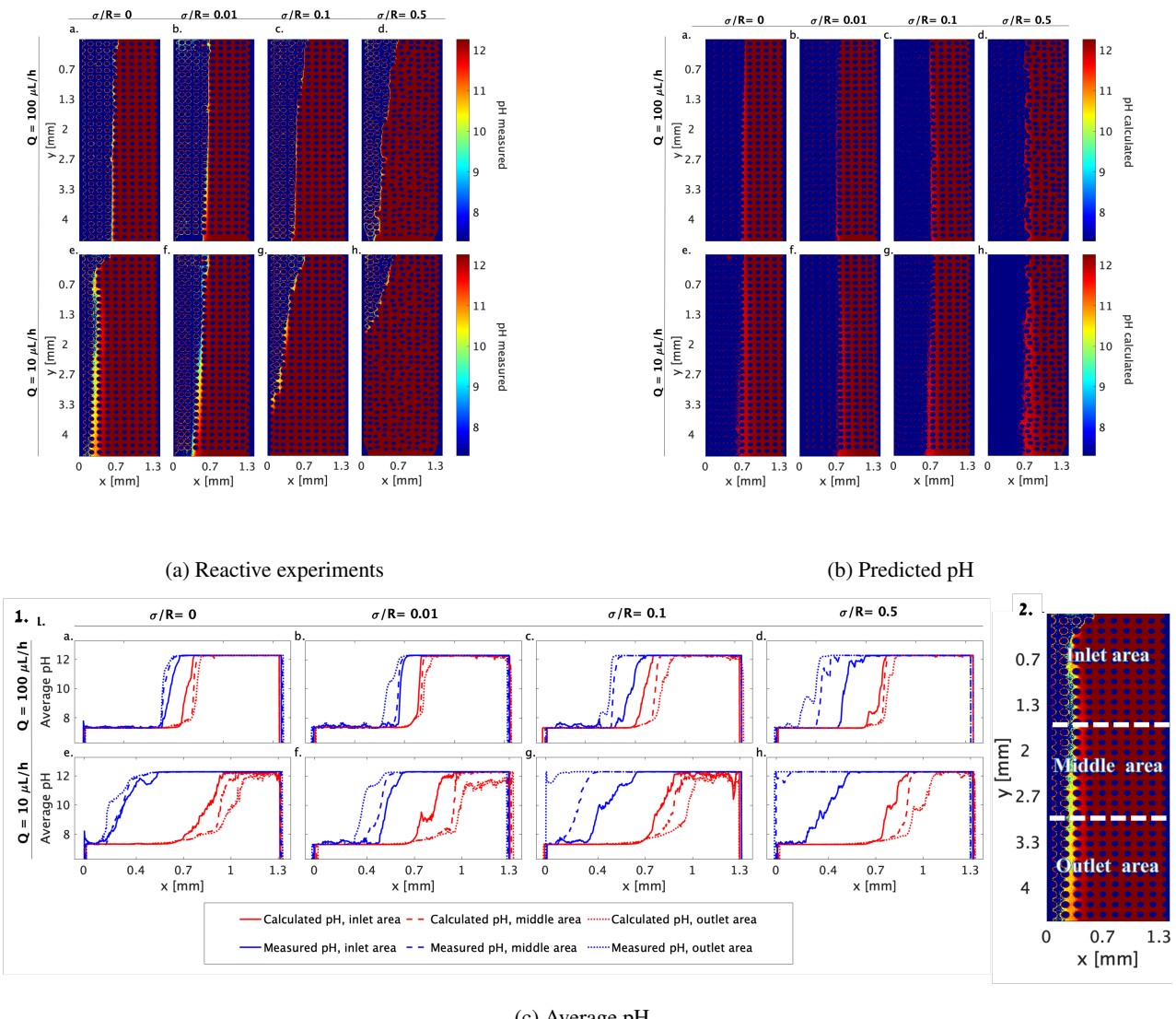

(a) Reactive experiments

(b) Predicted pH

(c) Average pH.

**Figure 4.** (a) pH values as indicated by the pyranine intensity distribution, (b) pH calculated from the normalized R6G concentration received in the mixing experiments using equation (8) for: a.-d. flow rate of 100 $\mu$L/h at medium heterogeneities of $\sigma/R = 0$, $\sigma/R = 0.01$, $\sigma/R = 0.1$ and $\sigma/R = 0.5$, respectively; and e.-h. flow rate of 10 $\mu$L/h at the marked medium heterogeneities, respectively and (c) 1. The corresponding pH averaging of (a) and (b) along the x-axis at three different sections of the flow cell (illustrated by the three, size-equal sections in (c) 2.): inlet area (continuous line), middle area (dashed line), and outlet area (dotted line), as calculated by the R6G concentration received in the mixing (red line) and as measured in the reaction (blue line).

along the x-axis. The plotted results, shown in Figures 4c.a.-h., clearly show how the transition in pH has a sharp interface for both the calculated and measured pH, even when averaged spatially; yet they also emphasize that while the calculated pH is

symmetric around the cell center, the experimental pH is very non-symmetrical and deviates significantly from the cell center, and this deviation between the calculated and experimental pH is worsened as the flow rate decreases.

The pH calculated from the mixing predicts that the average pH of the first section of the medium (red continuous line) starts to rise roughly in the middle of the cell. As we look at areas closer to the cell outlet, the average calculated pH (red dashed and dotted lines) starts to rise farther from the middle of the cell. However, the average pH measured in the reactive experiments

shows a different tendency - the farther away from the cell inlet, the higher the pH value is along the x-axis.

Moreover, the pH calculated from mixing predicts a more moderate climb of the average pH as the flow rate descends from 100 $\mu$L/h to 10 $\mu$L/h, which we do see in the pH spread measured in the reactive experiment, particularly in the inlet area (blue continuous lines in Figures 4c.1. top vs. bottom figure row). However, at the lower flow rate, the pH spread starts significantly closer to the inlet in all the tested heterogeneities. This points to a diffusion-related mechanism that is scaled between the high

to the low fluxes, following the Peclet relation in (2), yet this effect is dominant for the pH spread and not the R6G mixing pattern with the DDW.

The diffusion dominance is especially apparent in the spread of basfied pyranine at the expense of acidified pyranine for the lower flux, marking the rapid spread of low pH towards the high pH, yet the enhanced shear forces due to increased heterogeneity continue to play a role even in this low flux. Furthermore, the effect of heterogeneity on pH spread is also

reflected in the rapid spread of the middle cell baseified pyranine on the expense of acidified pyranine as the heterogeneity level increases, and with it, so are the shear forces. So, overall, although the neutralization reaction occurs faster than the R6G concentration gradient equilibrium, it appears to be affected by both the fluid flow rate and medium heterogeneity in a manner similar to that of the R6G.

### 3.2.2 The role of diffusion in pH transverse migration

While the logarithmically high OH$^-$ concentration explains the sharp pH change, the rate of migration, which breaks the symmetry between R6G and the acidify-basified pyranine pH measurement, follows the high proton mobility in water (Agmon (1995)). It has already been well established that proton transfers are one of the fastest chemical processes, and even in the diluted solutions phase, where diffusion is limited, their rates exceed other known reactions (Donten et al. (2012)). This is usually explained in terms of a sequence of proton-transfer reactions between water molecules along a hydrogen-bonded

network, known as proton hops, as described in the Grotthuss mechanism 200 years ago (Agmon (1995); Hassanali et al. (2011); Wolke et al. (2016)). Due to its tiny ionic radius and its strong polarization power, the proton cannot be isolated in equilibrium conditions. Instead, it immediately binds to an intact water molecule to form hydronium ions by creating covalent bonds (Thabet et al. (2020)).

The Grotthuss mechanism was proposed to explain how the excess proton occurring as hydronium ion diffuses much faster

than expected from its hydrodynamic radius. In this mechanism, the excess proton diffuses with a proton transfer from the hydronium to the neighboring water molecule, or from a water molecule to a neighboring hydroxide (Hassanali et al. (2011)).

The differences between the proton/hydronium diffusion rate to the R6G diffusion rate are reflected in their diffusion coefficients in water, as the former is more than one order of magnitude larger than the latter, with a diffusion coefficient ($D$) of

$9.3 \times 10^{-5}\,\mathrm{cm^2/s}$ (Amdursky et al. (2019); Zhang et al. (2021); Tuckerman et al. (2006)) for hydronium vs. $4 \times 10^{-6}\,\mathrm{cm^2/s}$ (Gendron et al. (2008)) for R6G, and close to one order for the OH$^-$, shown in Table 2.

| Chemical species | Diffusion [cm$^2$/s] | Ionic mobility [cm$^2$/sV] | Pe [100/10 $\mu$L/h] |
|---|---|---|---|
| H$^+$ | $9.31 \times 10^{-5}$ (1) | $36.2 \times 10^{-5}$ (2) | 7.6/0.76 |
| OH$^-$ | $5.27 \times 10^{-5}$ (1) | $20.6 \times 10^{-5}$ (2) | 13.7/1.3 |
| Cl$^-$ | $2.01 \times 10^{-5}$ (1) | $7.91 \times 10^{-5}$ (2) | 35/3.5 |
| Na$^+$ | $1.33 \times 10^{-5}$ (1) | $5.2 \times 10^{-5}$ (2) | 53/5.3 |
| Pyranine | $1.5 \times 10^{-5}$ (3) | | 47/4.7 |
| R6G | $0.4 \times 10^{-5}$ (4) | | 178/17 |

**Table 2.** Mapping of the various chemical components in our system with their corresponding diffusion, ionic mobility, and Peclet value for both experimental fluxes. Details can be found in: (1) (Parkhurst and Appelo (2013)), (2) Varcoe et al. (2014), (3) Himmelsbach et al. (1998), and (4) Gendron et al. (2008).

This high diffusion rate leads to a diffusion dominated transverse flux captured by the pH enhanced spread as the applied flux reduces, forming a low Pe over the pore size. Calculating the OH$^-$ transverse migration from the diffusion mean square displacement over the 10 s it takes for the fluid to advance the length of the cell (4.5 mm) for the 10 $\mu$L/h flow rate (recall that the $v_d = 0.0142\,\mathrm{cm/s}$), the high diffusion advances the OH$^-$ 0.2 mm. As diffusion is isotropic in nature, it not only occurs transversely to the flow but also aligned with the flow, leading to a steady state of OH$^-$ neutralized by the lower pH and marked by the acidified pyranine, as seen in the homogeneous case (Figure 4c.e). Multiplying this diffusion advancement by the Pe ratio reported in Table 1, brings this diffusion spread to 0.3 mm (see Table 3), nearly covering the full extent of the cell, and similar to the spread in Figure 4c.e. However, for the same extent of time and Darcy velocity, the high shear in the heterogeneous case further mixes the OH$^-$, leading to full homogenization of the pH in the flow cell (see Figure 4c.h, and Table 3). The shear increase with heterogeneity is marked by the reduction in the measured permeability presented in Table 3, which for fixed flow rate leads to higher shear. Yet, the same increase in shear between the homogeneous and the heterogeneous case for the high flux/Pe, produces a smaller relative effect on the OH$^-$ migration (Figures 4c.a-d).

| $\sigma/R$  [$-$] | 0.0 | 0.01 | 0.1 | 0.5 |
|---|---|---|---|---|
| $v_d$  [$mm/s$] | 0.146 | 0.155 | 0.155 | 0.162 |
| $MSD$  [$mm$] | 0.2 | 0.19 | 0.19 | 0.19 |
| $k \times 10^{-6}$  [$mm^2$] | 69 | 40 | 34 | 13 |
| $MSD \times Pe_{eff}/Pe$  [$mm$] | 0.297 | 0.31 | 0.336 | 0.468 |

**Table 3.** The table depicts the Darcy velocity, MSD, and MSD scaled by the effective Peclet ratio for each heterogeneity.

### 3.2.3 Calculating the ion concentration contribution to transverse migration

While the enhanced diffusion due to the Grotthuss mechanism is well established, the ionic nature of the proton/hydronium has the potential to form an electrical gradient that may affect the pH distribution. The contribution of this mechanism has been debated in the literature in the context of pH reaction, and shown that it can be neglected for brine, where the ion ratio is minor compared to the background solution (Li et al. (2006); Lichtner and Kang (2007); Li et al. (2007b)). Simulations performed on experimental results from a Hele-Shaw cell (Huang et al. (2023); Almarcha et al. (2010)) showed the importance of considering the Coulombic interaction via the Nernest-Plank equation and species-specific diffusion coefficient for density-driven flow in bulk. Additional simulations and experiments on porous media, coupling the electrical gradient with various flow configurations while considering the different diffusion values for each reactive species, yet for high Peclet, pointed to the important role of Coulombic interactions during reactive transport (Rolle et al. (2018, 2013)), yet it remains to be seen how relevant it is to the system presented in this study. yet it remains to be seen how relevant it is to the system presented in this study.

Following the Nernst-Planck equation (Bockris et al. (2006)), the flux of ions due to both diffusion and migration under an electric field is given by:

$$J_{OH^-} = -D_{OH^-} \frac{dC_{OH^-}}{dx} + u_{OH^-} C_{OH^-} \cdot E = -D_{OH^-} \frac{\Delta C_{OH^-}}{\Delta x} + u_{OH^-} C_{OH^-} \cdot E \tag{12}$$

where $J_{OH^-}$ [mol/cm$^2$·s] is the flux of the ion, $C_{OH^-}$ [mol/cm$^3$] is the concentration of the ion, $u$ [cm$^2$/sV] is the ionic mobility, and $E$ [V/cm] is the electric field. For the OH$^-$, the $D_{OH^-} = 5.3 \times 10^{-5}$ [cm$^2$/s], the $u_{OH^-} \approx 20 \times 10^{-8}$ [cm$^2$/(V·s)] (see Table 2 for details and reference), and knowing the concentration at both inlets and their distance ($\Delta x = 0.0475$ cm), the $E \approx -2.5693$ [V/cm].

We repeated this calculation for the hydroxide ions (OH$^-$), the protons (H$^+$), the Cl$^-$ and the Na$^+$, and for all cases, the diffusive flux (marked by the first term in (12) and scales with $D_{OH^-}$ [cm$^2$/s]) was two to three orders of magnitude greater than the electric flux (marked by the second term in (12) and is approximately $J_E(OH^-) \approx 2 \times 10^{-7}$ [mol/cm$^2$·s]) due to the ion concentration, making them negligible for our study (see Bard and Faulkner (2001) for details).

### 3.3 Combined Experiments

In this study, we identify this neutralization reaction effect by setting fluids with the same reactive tracer concentration, so a concentration gradient is not present for the pyranine, only the pH gradient. To mimic the practice where pH indicators are locally introduced and allowed to diffuse according to their concentration gradient and flow, we perform combined experiments where the pyranine is introduced only with the high pH inlet, similar to the basefied pyranine, yet it will need to migrate towards the low pH area where only DDW is present (acidified DDW). The conversion of image intensity to pH is performed as in the experiments in 3.2. In this setup, the basefied pyranine dispersion is the limiting reactant for the pH reaction, and as such, the role it has as a pH indicator is limited, as its dilution also acts as the limiting fluorescing factor, as shown in Figure 5.

Although we should expect the same pH distribution within the porous media as for the R6G test, given that the basefied pyranine diffusion is between the R6G and the OH$^-$ value (see 2), the image analysis yields a different distribution than the one

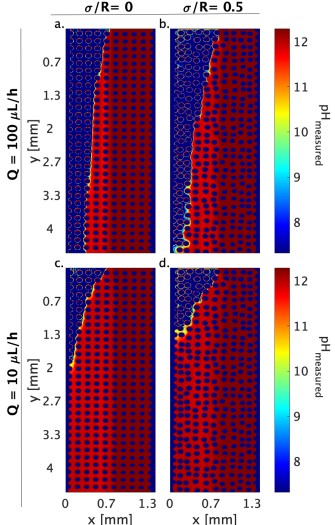

**Figure 5.** Combined experiments (both concentration gradient and pH gradient): pH values as indicated by the reactive tracer distribution, neglecting tracer concentration gradient. a. and b. Flow rate of 100 $\mu$L/h at medium heterogeneities of $\sigma/R = 0$ and $\sigma/R = 0.5$, respectively. c. and d. Flow rate of 10 $\mu$L/h at medium heterogeneities of $\sigma/R = 0$ and $\sigma/R = 0.5$, respectively.

accepted for the R6G or the acidfied/baseified pyranine (Figures 3 and 4a, respectively). While the pH change is only due to the occurring reaction, here, it is also wrongly accepted as a measure of the pyranine's transverse diffusion/dispersion, leading to the pyranine dilution, which can be wrongly accounted as the pH change. This is reflected in the measured pH of $\sim 11.5$ we see in Figures 5.a.-d., which does not appear in Figure 4a, and somewhat resembles the interface between the fluids around pH $\sim 11.5$ and pH $= 12.3$ in the pH pattern calculated from the R6G experiments.

Another interesting aspect is the fast migration of pyranine, which cannot be accounted for by pyranine diffusion. Calculating the transverse diffusion for the pyranine in the form of mean square displacement presents higher pH homogenization for the homogeneous case, than the case where pyranine concentration is uniform in the cell, raising the question of if the $OH^-$ concentration difference is facilitating migration in the form of osmotic pressure, while fixing the pyranine excitation levels.

### 3.4 Simulating the tracer and reactive experiments

The 2D tracer and pH experiment results in 3.1 and 3.2, respectively (for convenience, presented again in Figures 6.1. and 6.2., respectively), were reproduced using a COMSOL Multiphysics simulator (Figure 6.3 and Figure 6.4, respectively). This was done by importing the cell AutoCAD design to COMSOL so an exact Stokes flow simulation could be employed to solve the flow field, as described in section 2.3. The same parameters for the diffusion of the R6G and Darcy velocity were taken, and a forward solver was used to calculate the transverse dispersion. The simulation reproduced the initial invasion scheme for the experimental setup, and as in the experiment, the steady state was verified by comparing the output of consecutive time frames

in the simulation. Comparing Figure 6.1 with Figure 6.3 shows that the simulation captures the transverse dispersion for all Pe values well, although it somewhat enhances the dispersion for the low concentrations.

As in the R6G experiment, we use the same Stokes flow solver for the reactive experiment and update the diffusion coefficient to the higher value of the $OH^-$, and the $H^+$ migration (Table 2), providing the pH value based on both ion and proton diffusion. As the pyranine emission intensity range is given by (9), and shown by the exponential decay in Figure 2.b., a similar exponential conversion was used in the COMSOL simulation to highlight this region. Comparing the experimental and simulated reactive case (Figures 6.2 and 6.4, respectively) shows that the high Pe simulation produced similar results to the experimental values, yet the low Pe values proved more challenging. This difference between the simulation and experimental results, even when considering the pH exponential range, did not improve when we considered the Nernst-Planck equation ((12)). We believe that the discrepancy stems from the fact that while diffusion is indeed higher, the neutralization reaction of the $OH^-$ ions, and $H^+$ proton, following their local concentration, requires a model that also considers both the diffusion and local concentrations. Another aspect that is not considered in the COMSOL model is the Grotthuss mechanism and the strong polarization power leading to the immediate binding to an intact water molecule to form hydronium ions by creating covalent bonds, which is not considered in the enhanced diffusion directly. This mismatch between the model and the experimental setup clearly points to the need to incorporate these mechanisms in existing models.

## 4   Summary and Conclusion

### 4.1   Summary

We experimentally investigated the effect of porous media heterogeneity and flow rate on transverse mixing and their effect on the pH-driven neutralization reaction. The experiments showed that transverse mixing is controlled either by diffusion or by shear forces, while the former corresponds to the homogeneous medium and lower flow rate, and the latter corresponds to the heterogeneous medium and higher flow rate. Subsequent experiments followed the same flow rates and heterogeneity levels yet with a pH reactive tracer which provided the pH transverse dispersion. These pH reaction pattern does not necessarily follow the pH calculated by the mixing as the medium becomes more heterogeneous and the flow rate descends, and we trace this mismatch to the enhanced diffusion of pH. Another set of experiments were used to show how the measured pH can be wrongly interpreted once the tracer is not introduced uniformly in the domain. We simulated our system and show that the mixing experiments matched flow simulations, yet the simulated pH experiments with the enhanced diffusion capture the trend of pH transverse advancement, but not the local pH values.

### 4.2   Conclusion

The main conclusion from our experiments is that the pH gradient between the co-flowing fluids tends to equilibrate faster than the concentration gradient, so the reaction occurs earlier than predicted by the mixing pattern. The experiments demonstrate that the transition of a proton is considerably faster than that of diffusion and shear forces governing mixing. This can be

accounted for by several mechanisms, resulting in the abnormally high proton mobility in water, known as the Grotthuss mechanism. The experiments presented here show how important it is to consider when incorporating pH-driven reactions in porous media. Even so, diffusion alone is not sufficient when considering neutralization reactions like pH, as is clear from the mismatch between the COMSOL simulations and the experimental results.

This difference in diffusion rate can be easily missed as in most experimental setups, and in the field, pH is generally measured locally or with a pH indicator that migrates with the flow. The pH may equilibrate faster than the pH indicator diffusion due to the hydronium ions binding transition that does not require movement of the ion, only dissociation of the water, while the pH indicator and/or reactants should distribute slower in the porous media as they lack this mechanism. Moreover, to avoid the charged balance calculation between the various ions and cations in the reactive system, with their respective diffusion coefficients (Table 2), studies often assume that diffusion is uniform for all chemical species. This assumption may hold while the background salinity is high (namely, close to seawater), yet for low salinity water, this assumption becomes questionable (Lichtner and Kang (2007); Li et al. (2007a); Lichtner (1996)).

As pH reactions are the most frequent and abundant reactions in soil and rock formation, considering the differences between pH migration and mixing is crucial to capturing the extent of reactions. Our findings raise questions on the assumption that the diffusion differences between chemical species, specifically for pH reactions but also for various chemical species, as evident from the difference of transverse migration of the R6G and pyranine, are negligible. This assumption may be valid for higher Peclet, or for specific chemical species, yet as typical flows in soil and rock generally follow low Peclet and involve pH reactions and/or rich ion composition, this assumption is rarely true.

**Code and data availability**

Code and data are available on a dedicated GitHub repository upon request to Yaniv Edery (yanivedery@technion.ac.il).

**Author contributions**

AB developed the experimental methodology and performed all the experiments, analyzed data, and wrote major parts of the paper. TS developed the simulation code and performed numerical simulations. LA helped develop the experimental methodology. YE helped develop the experimental methodology, supervised and guided the experiments and simulations, and wrote major parts of the paper.

**Competing interests**

The contact author has declared that neither of the authors has any competing interests.

**Financial support**

This research has been supported by the German-Israeli Foundation (grant no. I-2536-306.8), and the Israel Science Foundation (grant no. 801/20).

**Acknowledgments**

YE and AB thank the German-Israeli Foundation (grant no. I-2536-306.8). YE, TB, and LA thank the Israel Science Foundation (grant no. 801/20).

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

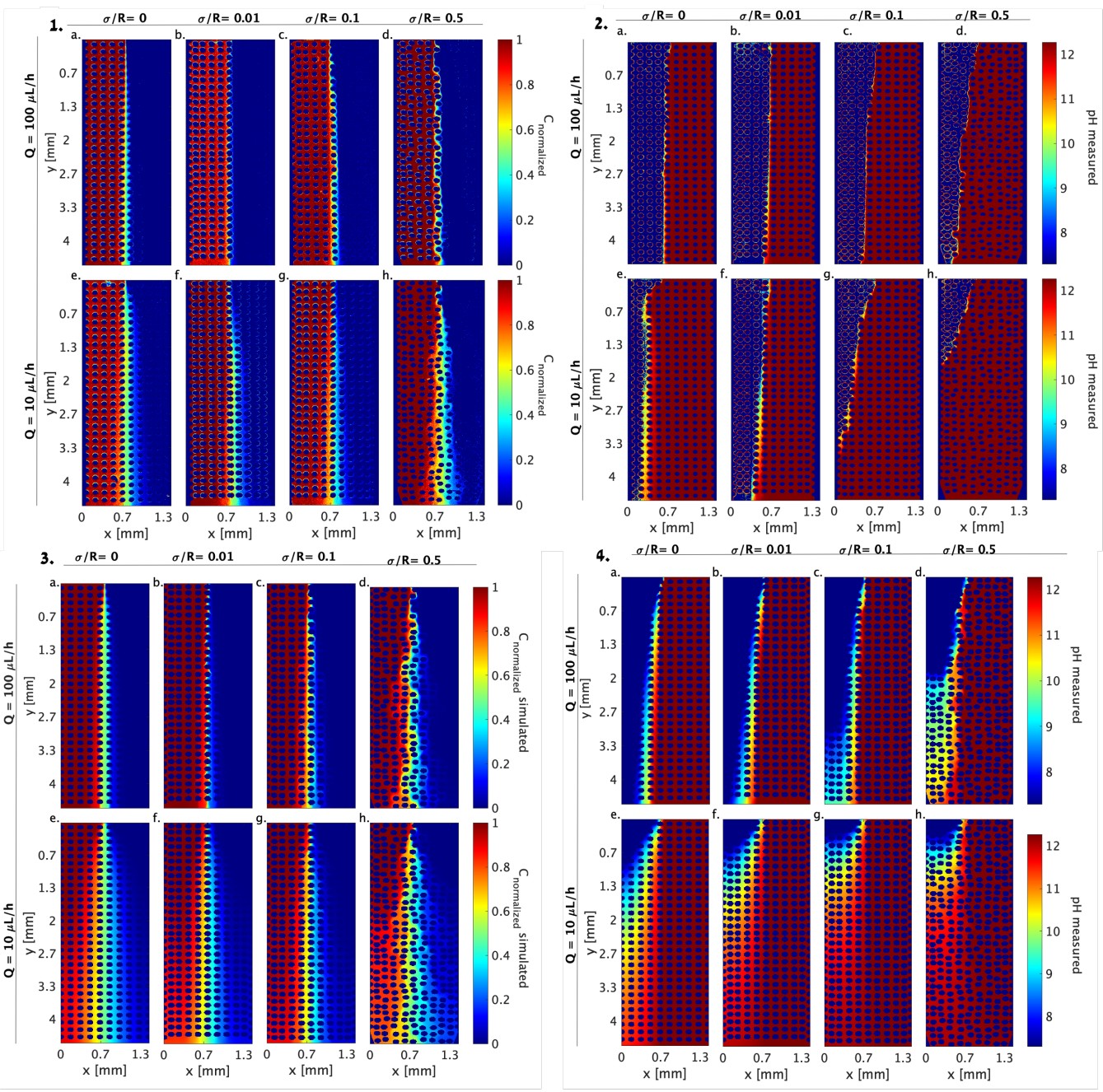

**Figure 6.** Distribution of 1. R6G normalized concentration, 2. pH measured by the pyranine, 3. Simulated R6G normalized concentration, and 4. Simulated pH, provided for: a.-d. Flow rate of 100 $\mu$L/h, and e.-h. Flow rate of 10 $\mu$L/h at presented medium heterogeneities. The 2D $R^2$ for the R6G simulations was above 0.88 for all hetrogeneities and flowrates, and 2D $R^2$ values ranging from 0.6 to 0.75 for the pH simulations, where the high heterogeneity and low flowrates portray lower 2D $R^2$