# Peer review of "Experimental investigation of the interplay between transverse mixing and pH reaction in porous media"

_EGUsphere, 2024_

## Referee Comment (RC1)

**Review of the manuscript entitled "Experimental investigation of the interplay between transverse mixing and pH reaction in porous media" in the *HESS* journal**

**General comments**

The article proposes a research topic consistent with the journal. The article focuses on the impact of pH on mixing in porous media, using microfluidics for process visualization. The experimental method chosen is appropriate to the objectives, however, the numerical calculations and simulations proposed deserve some adjustments to improve the discussion of the results and the conclusions of the study.

The quality of the language used in the manuscript is good, but the writing needs to be adjusted to make it easier to understand. Indeed, although most of the elements required for comprehension are given, the use of ambiguous terms and the choices made in the article structure make reading more complex. In addition, the supplementary material contains "scientific interpretations or findings that would go beyond the contents of the manuscript" contrary to the journal guidelines. Restructuration of the manuscript must be done to include these numerical results.

**Specific comments**

1. It would be beneficial to achieve a more balanced distribution of the introduction between the general topic of reactive transport and the specific mixing issue. Additionally, the reactive transport section is somewhat difficult to follow due to the frequent shifts in scale.
2. At the end of the introduction, conclusions from the observation of the results are presented. However, the reader expects a summary of the plan of the article. The actual presentation of the plan could be more detailed than in its present form.
3. The use of "pH reactant", "tracer", and "background solution" to designate the different solutions used in the experiments is confusing. It would be clearer for the reader to use only specific designations, such as "R6G", "acidified pyranine", "basified pyranine", and "basified DDW". This is also the case of the legend of the scheme in Figure 1a, which would then be homogenized with the legends 1b and 1c. In addition, the left and right sides explain from which side the tracer or the background solution is injected. However, this orientation is reversed in Figure 1 compared to the description in the text and the results. In the same way, the designation of "mixing" or "reactive" is confusing because there is mixing in all of these experiments. This could be replaced by "R6G and DDW mixing", "acidified and basified pyranine mixing", and "acidified pyranine and basified DDW mixing".

4. Some information about the experimental setup is missing in Section 2.1: the syringe pump model, flow rate values, density and salinity of the solutions, and temperature. Flow rates are indicated later in Section 2.2, but mentioning them first would be more logical when presenting the setup.

5. Some information about the imaging is missing in Section 2.2: the model of the microscope and the timeframe for the captured image sequence. In addition, it is unclear if all the presented images were taken after waiting 5 or 10 minutes for the respective flow rates of 100 and 10 µL/h or from the second capture (at 10 or 20 minutes, respectively) or if a mean image was created. The authors refer to image sequence acquisition and two acquisition periods, but the images' quality and stability over time are never discussed in the presentation of the results.

6. There is a numbering problem with the subsection "Image Analysis", which could be merged with section 2.2 as "image capture and analysis". In addition, the following subparts could be renamed not following the "mixing" or "reactive" designation, but tracer type.

7. The calibration curve for pyranine is obtained at a constant concentration. In the experiments of acidified pyranine and basified DDW mixing, the authors refer to the dilution of pyranine to explain the difference in pH observed compared to the results obtained with the acidified and basified pyranine mixing. However, pyranine is injected from the left side, and from the pH values, no dilution is observed on the opposite side of the cell with pH at 12.3, only in the middle with a lower pH value and brighter red color. Did the authors consider different calibration curves from lower pyranine concentrations?

8. The equations (7), (8), and (9), as well as the features of COMSOL simulations should be presented in the Method section. It is unclear if concentration $C_{ij}$ refers to pyranine of R6G value in equation (8), the equation to calculate the pH. The equation to obtain the electric field value should also be added. The reference to $J_D$ and $J_E$ should be clarified in equation (9), the equation of the ion flux J.

9. There is no pH contrast in the experiments of R6G mixing. Figure 3.2 comes from synthetic results. This should be clearly stated and since this figure has to be compared with Figure 4, it would be more logical to merge it with Figure 4 and not describe it in section 3.1, but in section 3.2 instead. In addition, the results obtained in Figure 3.2 are surprising. From the color scale, no intermediate pH is observed at the mixing boundary (no visible yellow or green transition). I hope the authors can comment on this.

10. All the results of mixing experiments involving pH contrast show alkaline invasion in the left half of the cell. I hope the authors can discuss this effect regarding the initialization procedure, which consists of saturating the cell first with the alkaline solution.

11. Line 219, the thickness and symmetry of the interface are discussed based on Figures 3.2 and 4. Aside from the fact that this part of the discussion is difficult to follow, the interface is more visible in Figure 5. I suggest some discussion reformulation based on this.

12. The average pH along the x-axis at three different sections of the cell of the experiment of acidified pyranine and basified DDW mixing could be added to Figure 5 and discussed in a separate subsection 3.4.

13. The solutions used in the experiments have different chemical compositions, thus certainly different salinities, viscosities, and densities, affecting flow and mixing. I hope the authors can provide the information and add some comments to the discussion.

14. I am surprised by the methodology of calculating a specific Péclet number for each species, rather than calculating effective diffusion coefficients to obtain the Péclet numbers of each solution. Comparing the effective diffusion coefficients would bring more value to the discussion when comparing the results from the experiments of acidified and basified pyranine mixing with acidified pyranine and basified DDW mixing.

**Technical corrections**

- The abstract is usually a single paragraph.
- There is a missing space in the text before the left parenthesis of all citations.
- Two citations in the same sentence should be grouped in the same parentheses (e.g., line 21).
- Line 27: the formulation "geological studies, to watershed or global issues" could be clarified.
- The units are not presented in the SI format (e.g., "µl" should be "µL", "hr" should be "h", "sec" should be "s") and the fraction format should be avoided and replaced with the inline format.
- The captions of figures 3 to 5 could be shortened since medium heterogeneities are already labeled.
- Figure 1e: the curves cannot be distinguished based on the chosen colors in the legend, even for non-color-blinded readers.
- There is a numbering problem for the equation of the ion flux: (8) should be (9).
- Line 269: misspelled "relay" should be replaced by "rely".
- Units are missing when presenting the ionic mobility and the electric field (line 300), then for $J_D(OH^-)$ and $J_E(OH^-)$.
- The guidelines recommend separating the summary from the conclusions.

---

## Referee Comment (RC2)

The manuscript provides very interesting experiments and interpretations.

However, it requires revision before publication in HESS for the following reasons:

- It should contain a state of the art of the reactive transport experiments already performed in the past and better highlight the contribution of this paper (see for example additional paper listed below and references here in).
- The supplementary material should be included in the manuscript. The processes considered in the modelling should be detailed. Parameters, boundary conditions, space and time discretizations for the simulation of the flow and the transport should be given and justified. The differences between the model and the experimental data should be commented in details.
- The concentration normalization assumes a linear relationship between intensity and concentration (eq. 5). Please comment.
- What is the reliability of an experiment, i.e., what is the difference in the transverse missing/dispersion when repeating a reactive transport experiment?
- I do not understand why the distance between the two inlets is used for the computation of the ion fluxes due to non-neutral electric field generated by differences in the diffusion coefficients (Nernst-Planck equation). This gradient should be a local one (not at the scale of the setup), that can be computed using the images. Please clarify and comment.

**Minor comments:**

What is the pixel size of the images? What is the reliability of the concentration values (0.1 % camera noise but what about the uncertainty related to the calibration)?

Please check units through the document and use SI format.

L108: "simultaneous" -> "simultaneous"

L269: "relay" -> "rely"

L133: a character space is missing "solution(Barzan and Hajiesmaeilbaigi, 2018)". Same for L140, L194.

Two "=" in equation (8) which should be (9).

**References**

Huang et al., 2023, doi.org/10.5194/gmd-16-4767-2023

Loyaux-Lawniczak S. et al. (2012) DOI: 10.1016/j.jconhyd.2012.06.005

Rolle et al., 2013. doi.org/10.1016/j.gca.2013.06.031

Liu, C. et al., 2011, doi.org/10.1029/2011WR010575

Rolle et al.,2018, doi.org/10.1002/2017WR022344

---

## Author Comment (AC1)

**Review of the manuscript entitled "Experimental investigation of the interplay between transverse mixing and pH reaction in porous media" in the *HESS* journal**

**General comments**

The article proposes a research topic consistent with the journal. The article focuses on the impact of pH on mixing in porous media, using microfluidics for process visualization. The experimental method chosen is appropriate to the objectives, however, the numerical calculations and simulations proposed deserve some adjustments to improve the discussion of the results and the conclusions of the study.

The quality of the language used in the manuscript is good, but the writing needs to be adjusted to make it easier to understand. Indeed, although most of the elements required for comprehension are given, the use of ambiguous terms and the choices made in the article structure make reading more complex. In addition, the supplementary material contains "scientific interpretations or findings that would go beyond the contents of the manuscript" contrary to the journal guidelines. Restructuration of the manuscript must be done to include these numerical results.

We thank the reviewer for recognizing the quality of our work and for identifying its relevance to the HESS readership. The comments from the reviewer were extremely helpful in clarifying ambiguous terms and improving the paper structure and were pivotal in addressing the simulations and calculations in our study. These comments also aided in improving the layout of our figures and how they are addressed in the manuscript. We are grateful to the reviewer for the time invested in reviewing our work. The reviewer's comments are marked in black, while our replies are marked in blue. We believe that we have addressed all the comments.

**Specific comments**

1.      It would be beneficial to achieve a more balanced distribution of the introduction between the general topic of reactive transport and the specific mixing issue. Additionally, the reactive transport section is somewhat difficult to follow due to the frequent shifts in scale.

We thank the reviewer for suggesting ways to improve the readability of our manuscript. The imbalance in the introduction stems from the fact that while pH-induced reactions are abundant in soil processes, they are considered a sub-subject of the reactive transport process in porous media and share many of the physical mechanisms in it. Therefore, it is important to acknowledge the vast work done in reactive transport, yet we agree that a balanced introduction will better serve the manuscript. We therefore condensed the section dealing with the general

aspects of reactive transport and mixing so to emphasize the pH spread and pH reactions. Specific changes were made in line 25 and lines 35-40 which were extracted (marked below) while their references were moved.

"It is important for the understanding of numerous Earth Sciences problems ranging from engineering applications such as carbon capture and storage or groundwater remediation, as well as geological studies, to watershed or global issues(Carrera et al., 2022)."

"At the macroscale, the advection-dispersion reaction equation (ADRE) is usually used to describe reactive transport. However, it may provide incorrect predictions of experimental results in reactive systems, including the extent of reactions in mixing-controlled chemical transformations(Battiato and Tartakovsky, 2011; Berkowitz et al., 2016). This is mainly because the ADRE is not sensitive to incomplete mixing at the pore scale, in which biogeochemical reactions occur(Edery et al., 2013, 2009; Alhashmi et al., 2015). "

2.      At the end of the introduction, conclusions from the observation of the results are presented. However, the reader expects a summary of the plan of the article. The actual presentation of the plan could be more detailed than in its present form.

We thank the reviewer for proposing this summary at the end of the introduction. It was indeed needed as a segue for the following sections and should have been there in the first place. We have introduced this summary starting in line 54 (now line 40) with a sentence that marks the need for the experiments we performed, followed by a summary of the rationale behind them. We proceeded by altering the following sentences to detail the plan for the rest of the paper, as requested.

"Considering the coupling between mixing and reactive transport processes and how both are scaled with the heterogeneity, specifically in the context of pH reactions in heterogenous soil, a set of experiments is proposed to observe if, indeed, the same coupling between mixing and reaction occurs for pH spread and reactions. These experiments focus on investigating how porous medium layouts ranging from homogeneous to heterogeneous affect pH-driven reactions by examining the pattern of transverse dispersion of co-flowing fluids for both mixing and pH. This is done by tracking the mixing and pH spread for two Peclet values using fluorescently labeled fluids imaged by a confocal microscope. The mixing experiments showed that transverse mixing varies from diffusive mixing in the homogeneous case to shear-driven mixing in the heterogeneous case. However, the pH measured in the pH experiments does not follow the pH value calculated from the mixing pattern. Instead, it shows a larger spread, suggesting that the co-flowing fluids' pH difference equilibrates faster than the mixing. We identify the proton transfer mechanism, which is comparatively faster than the transverse dispersion or diffusion, as the dominant mechanism, especially for lower Peclet. Pore-scale

simulations agreed well with the mixing experiments and provided reasonable results for the pH experiments after considering the enhanced diffusion due to the proton transfer mechanism."

3. The use of "pH reactant", "tracer", and "background solution" to designate the different solutions used in the experiments is confusing. It would be clearer for the reader to use only specific designations, such as "R6G", "acidified pyranine", "basified pyranine", and "basified DDW". This is also the case of the legend of the scheme in Figure 1a, which would then be homogenized with the legends 1b and 1c.

We agree with the reviewer that unifying the terms for the fluid solutions in our paper is important. We have altered the necessary terms, both in the text and figure caption. We have kept the terms "mixing experiments," "reactive experiments", and "combined experiments" since they signify the main process underlining the experiment. This has undoubtedly improved the manuscript's clarity. Thank you for this suggestion.

In addition, the left and right sides explain from which side the tracer or the background solution is injected. However, this orientation is reversed in Figure 1 compared to the description in the text and the results.

Thank you for catching this discrepancy between our figures, and between the figures to the text. As requested, we have modified Figure 1 to align with the description in the text and the results.

In the same way, the designation of "mixing" or "reactive" is confusing because there is mixing in all of these experiments. This could be replaced by "R6G and DDW mixing", "acidified and basified pyranine mixing", and "acidified pyranine and basified DDW mixing".

While we agree with the reviewer that unifying the terms for the fluid solutions in our paper is indeed important, we are not so sure regarding the mixing and reactive terms. Indeed, as the reviewer stated, the mixing occurs for all these experiments, yet for the R6G, it is the mixing between the R6G and the DDW, while for the reactive experiments, it is the pH mixing, or more accurately, the pH spread as the Grouthoss effect considers both the diffusion and ion polarization effect. Therefore, while the pH spreads, the pyranine reacts with the $OH^-$ in response to this pH spread. As such, we cannot call it acidified and basified pyranine mixing, as it is the acid and base that mix and not the pyranine. We have tried to be more specific regarding which process is occurring if it is mixing between R6G with the DDW in the mixing experiments or pH spread measured by the acidified and basified pyranine (now stated in key places in the manuscript).

4. Some information about the experimental setup is missing in Section 2.1: the syringe pump model, flow rate values, density and salinity of the solutions, and temperature. Flow rates are

indicated later in Section 2.2, but mentioning them first would be more logical when presenting the setup.

We appreciate the opportunity to correct these missing parts. We have inserted the needed fluxes and their corresponding Darcy velocities, as well as the Peclet equation and Peclet equation modified by the effective diffusion coefficient suggested by the reviewer, into section 2.1 and in a dedicated table. We also added the temperature and salinity values.

"Each cell had two parallel inlets (right and left), each of them set at 425 µm from the edge of the cell, and one funnel shaped outlet. At the two outlets, a syringe pump (Chemyx Fusion 200 Two Channel model) with a small diameter glass syringe (100 µL Hamilton glass syringe) allowed a continuous movement for the motor and the piston with no oscillations for the applied fluxes (100, and 10 µL/h flow rate, resulting in a Darcy velocity of $v_d$ = 0.142, and 0.0142 cm/s, respectively). These two velocities provided two Peclet numbers (Pe), as depicted by the following equation:

$$Pe = \frac{v_d R}{D} \qquad (1)$$

The Peclet number is a measure of the velocity magnitude ($v_d$), and the diffusion ($D$), which is an intrinsic property of the fluids over the mean pore size (R) (Bossis and Brady (1987)). While the mean pore size remains the same for all heterogeneity, there are small porosity ($\phi$) variations (see Table 1 for details). However, the main heterogeneity effect is on the interface between the co-flowing fluids, forming a torturous path. To address this, we define an effective diffusion coefficient $D_{eff} = \frac{D\phi}{T}$, which scales the diffusion of the reactants in water, as shown in many studies (Ray et al. (2018); Fogler (2011); Guo et al. (2022); Kim et al. (1987); Quintard (1993); Quintard and Whitaker (1993); Beyhaghi and Pillai (2011)). The tortuosity ($T$) can be directly calculated from the normalized standard deviation σ/R, which marks the range for the pillar center movement from a uniform grid using the following relation, $T = 1 + \sigma/R$, and leading to the effective Pe number of:

$$Pe_{eff} = \frac{v_d R T}{D\phi} \qquad (2)$$

and scaling the Peclet number as depicted in Table 1.

| $\sigma/R$ $[-]$ | 0.0 | 0.01 | 0.1 | 0.5 |
|---|---|---|---|---|
| $\phi$ $[-]$ | 0.68 | 0.64 | 0.64 | 0.62 |
| $T$ $[-]$ | 1 | 1.01 | 1.1 | 1.5 |
| $Pe_{eff}/Pe$ $[-]$ | 1.47 | 1.58 | 1.72 | 2.42 |

**Table 1.** The table depicts the porosity, tortuosity, and effective Peclet ratio for each heterogeneity.

The fluorescent conservative tracer used for the mixing experiments (Figure 1.a.) is rhodamine 6G (R6G), which is widely used to visualize flow patterns, such as in the domain of environmental hydraulics (Barzan and Hajiesmaeilbaigi (2018)). Pyranine (8-hydroxypyrene-1,3,6-trisulfonate)

is used for the reactive and combined experiments (Figure 1.b.-c.) as the pH reactant, as its fluorescent emission spectra and intensity are highly dependent on medium pH (Avnir and Barenholz (2005)), therefore suitable for monitoring pH changes. The R6G's concentrations were 2 mg/50 ml double distilled water (DDW with ≈ 18 MΩ·cm$^{-1}$ at 25 ∘C, the lab temperature, purified by Milli-Q) for the R6G (corresponding to 0.083 mM) and 9 mg/50 ml DDW for the pyranine (corresponding to 0.347 mM). These concentrations had no measurable effect on the fluid viscosity and density in this experimental setup."

[Figure]

**Figure 1.** a. An illustration of the mixing experiment setup. b. An illustration of the reactive experiment setup (pH gradient only). c. An illustration of the combined experiment setup (pyranine concentration gradient and pH gradient). d. Four different pore size variations (heterogeneities) of the flow cell, from the homogeneous one ($\sigma/R = 0$) to the most heterogeneous ($\sigma/R = 0.5$). e. Intensity of pyranine emission on a logarithmic scale versus wavelength for various pH, as measured by UV-vis, and verified under the confocal. The inset is a blow-up on a linear scale to present the relevant separation of the pH to intensity.

5. Some information about the imaging is missing in Section 2.2: the model of the microscope and the timeframe for the captured image sequence. In addition, it is unclear if all the presented images were taken after waiting 5 or 10 minutes for the respective flow rates of 100 and 10 μL/h or from the second capture (at 10 or 20 minutes, respectively) or if a mean image was created. The authors refer to image sequence acquisition and two acquisition periods, but the images' quality and stability over time are never discussed in the presentation of the results.

We thank the reviewer for pointing out the missing information and apologize for the obscure explanation. We have now included an explanation of the criteria for establishing image quality within the sequence.

"For the 100 μL/h flow rate, a series of 50 pictures were taken 5 minutes after forming a stable interface between the fluids. Then, after an additional 5 minutes of delay, another series of 50 pictures is taken, under the same conditions. The two series of images are compared to verify the stability of the interface. For the 10 μL/h flow rate, the same imaging sequence was performed, with an initial time of 10 minutes and a subsequent delay time of 10 minutes. For

both flow rates, each pixel intensity (marked as I$_{ij}$, for location ij) at each 50 pictures sequence, the variance of intensity per pixel did not exceed the 0.1% white noise of the camera. To verify that the interface among image sequences is stable, the criteria was set that the difference between the initial and later imaging sequence that exceeded the 0.1% (white noise of the camera) was averaged in absolute terms, and the stability of the interface was established if the average difference was isotropic and smaller than 1% (namely, , $\langle \frac{|I_{ij}(t=5)-I_{ij}(t=10)|}{|I_{ij}(t=10)|} > 0.1\% \rangle <$ 1%) a similar analysis was performed around the interface to verify that the 1% difference is not the outcome of the bulk behavior."

6.  There is a numbering problem with the subsection "Image Analysis", which could be merged with section 2.2 as "image capture and analysis". In addition, the following subparts could be renamed not following the "mixing" or "reactive" designation, but tracer type.

We apologize for this error. The merging is now fixed. However, we kept the "mixing" and "reactive" since it is in line with the result section titles.

7.  The calibration curve for pyranine is obtained at a constant concentration. In the experiments of acidified pyranine and basified DDW mixing, the authors refer to the dilution of pyranine to explain the difference in pH observed compared to the results obtained with the acidified and basified pyranine mixing. However, pyranine is injected from the left side, and from the pH values, no dilution is observed on the opposite side of the cell with pH at 12.3, only in the middle with a lower pH value and brighter red color.

   Did the authors consider different calibration curves from lower pyranine concentrations?

We are not entirely sure we followed the reviewer's comment. In the acidified pyranine and basified DDW experiments, there is no partitioning of the dilution from the reaction of the pyranine using a calibration curve dedicated to this aspect. The aim of the combined experiment is to show that there is a discrepancy between using the pyranine in one inlet and using it in two inlets. As such, there was no foreseeable reason to actually verify and separate the dilution from the reaction, nor did we manage to come up with an appropriate experimental methodology to do so. As the change in pyranine emission is due to pH, and the dilution of the pyranine manifests in the signal amplitude, there is no straightforward way to separate the two. The only way to qualitatively estimate the magnitude of dilution and pH change is through a comparison between the acidified pyranine and DDW-basified experiment with the acidified and basified pyranine experiment for similar conditions. However, we feel that this is out of the scope of this study, as it will require some assumptions and mainly simulations aimed at decoupling these two processes. We are currently working on generalizing the simulations we do have to other conditions, yet this is not in the scope of this current study.

The equations (7), (8), and (9), as well as the features of COMSOL simulations should be presented in the Method section. It is unclear if concentration $C_{ij}$ refers to pyranine of R6G value in equation (8), the equation to calculate the pH. The equation to obtain the electric field value should also be added. The reference to $J_D$ and $J_E$ should be clarified in equation (9), the equation of the ion flux J.

We indeed pondered what will be the best way to present the COMSOL simulation part of this work, mainly since we feel that the simulations are not seminal for the conclusions from this work. We are also working on generalizing our simulations to various cases, as stated previously, and the simulations in the paper are aimed at pointing toward the modeling gaps. Nonetheless, we accept this suggestion from the reviewer and have moved the simulation part to the main text, as well as updated the needed equations. We also clarified the $C_{ij}$ definition and updated the chapter on the electric field calculation using the Fokker-Planck equation, specifically addressing the needed clarifications on the fluxes. We have also moved equations 7 and 8, with their accompanying text, to the method section as they fit well with the context of the experimental setup. However, we are not sure that equation 9 fits there since the debate on the necessity to consider the Fokker-Planck equation is motivated by the fact that the diffusion term alone is insufficient in capturing the transverse spread of the pH. We, therefore, left it in the results section.

8.      There is no pH contrast in the experiments of R6G mixing. Figure 3.2 comes from synthetic results. This should be clearly stated and since this figure has to be compared with Figure 4, it would be more logical to merge it with Figure 4 and not describe it in section 3.1, but in section 3.2 instead. In addition, the results obtained in Figure 3.2 are surprising. From the color scale, no intermediate pH is observed at the mixing boundary (no visible yellow or green transition). I hope the authors can comment on this.

We agree with the reviewer that this should be compared with Figure 4 and not necessarily with Figure 3.1. We felt that they should be presented together since one is the outcome of the other, and that Figure 3.2 is clearly different from Figure 4. This obvious difference between the figures makes the comparison between them cumbersome, yet the observable difference that was noticed by the reviewer is important. This difference stems from the logarithmic relation between R6G concentration and pH value in equation 8 in the new manuscript. As such, there are, nor should there be, intermediate values, as the logarithmic transformation is not sensitive enough to present them. This is briefly explained in the text:

"The calculated pH at 10 µL/h flow rate (Figures 4b.e.-h.) predicts a somewhat asymmetrical pattern regarding the basified vs. acidified pyranine distribution, and a slightly narrowing strip of the basified pyranine as fluids move towards the outlet zone, indicating that the reaction theoretically gets larger due to R6G diffusion. This increase is the outcome of the logarithmic scale of pH (see (8)), where the molar value of the access $OH^-$ ion are orders of magnitude

higher on one side, which dominates over the cross-section. However, reactive experiments show that basified pyranine moves vertically along the cell significantly more than the calculated pH predicts, and the volume of the basified pyranine is increased at the expense of the acidified pyranine. This demonstrates that the reaction does not necessarily follow the mixing pattern in porous media, as the pH spreads faster than the R6G concentration predicts, an aspect that has a clearer representation in the following section. "

We therefore agree that this aspect requires a more elaborate explanation, and a better representation in the figures. We therefore added the calculated pH figure to be with the measured pH figure, and transformed Figure 4 to be Figures 4.1 and 4.2 instead of Figure 3.1 and 3.2, and we stressed that Figure 4.2 is indeed the product of Figure 3.1 and equation 8 in the caption and in the text. We further stress the logarithmic nature of equation 8 and how it leads to a sharper transition of the pH.

"This sharp interface in pH value is probably due to the pyranine intensity exponential decay (Figure 2), and the logarithmic scale at which concentration is transforming to pH in (8)."

Following the reviewer's subsequent comment, we also incorporated Figure 5 into the set of Figures 4.1 and 4.2, which is now Figure 4.3. This allows the reader to compare both the measured and calculated pH local spread with the averaged value.

9.      All the results of mixing experiments involving pH contrast show alkaline invasion in the left half of the cell. I hope the authors can discuss this effect regarding the initialization procedure, which consists of saturating the cell first with the alkaline solution.

The alkaline nature of the initial solution and spread is stated in the sentence we adjusted in response to the previous comment, and an additional sentence regarding the alkalinity appears at the beginning of section 3.2.1. (now 3.2.2.): "While the logarithmically high $OH^-$ concentration explains the sharp pH change, the rate of migration, which breaks the symmetry between R6G and the acidify-basified pyranine pH measurement, follows the high proton mobility in water (Agmon270 (1995))."

Regarding the initialization procedure, we initially saturate with the highest pH (12.3), and the lower pH is introduced subsequently, which has an influence on the first pore volume. However, a pore volume in our flow cell takes 6 and 60 seconds for the 100 and 10 ul/min, respectively. Therefore, we introduce between 20 to 100 pore-volumes until we take the images presented here. Furthermore, we compare the images as stated in section 2.2, and if there were an influence of the initialization scheme, we would have seen a consistent change between the subsequent measurements. We were afraid that the PDMS might act as a reservoir for the initial

fluid and change the pH, yet both the literature and the measurements we did, where we rinsed the PDMS in basified solution and then inserted it into a bath with the acidified solution (after wiping it lightly), showed no pH change in the solution as measured by pH meter.

We refer to this in the following sentence:

"This substantial migration of pH towards the acidified solution (recall that the pyranine concentration is uniform throughout the cell, and only the pH differs) cannot be the outcome of the initialization of solutions in the flow cell, as these measurements were taken after 100 and 20 pore-volumes for the 100 and 10 μL/h flow rates, respectively."

Furthermore, in the submitted version, there was a relevant sentence that commented on the migration of pH, which was wrongly placed in the methods section (lines 161-165), where the pH migration is less observable (figure 2) and lacks the right context. We now moved this sentence to the right place in the result section:

" In both the intensity data and the analyzed pH image, there is a noticeable transition of the interface from left to right, or from high pH to low pH. This transition is due to the diffusive nature of the $OH-$ions from their higher concentration to their lower concentration. This shift in the interface was reported in previous studies, leading to the shift in precipitation of $CaCO_3$.(Katz et al., 2011; Tartakovsky et al., 2008, 2007) As such, observing this shift in our experimental setup is in line with previous studies on pH induced reactions."

10.    Line 219, the thickness and symmetry of the interface are discussed based on Figures 3.2 and 4. Aside from the fact that this part of the discussion is difficult to follow, the interface is more visible in Figure 5. I suggest some discussion reformulation based on this.

We thank the reviewer for this comment, which helped us focus and articulate our findings better. Following this advice, we moved section 3.2.2 to 3.2.1, and focused the discussion on section 3.2. on the relation between the sharp pH transition and the symmetry of the pH. We also added this sentence to what is now section 3.2.1: "In each section, we calculated the average pH of each column of the matrix along the x-axis. The plotted results, shown in Figures 4c.a.-h., clearly show how the transition in pH has a sharp interface for both the calculated and measured pH, even when averaged spatially; yet they also emphasize that while the calculated pH is symmetric around the cell center, the experimental pH is very non-symmetrical and deviates significantly from the cell center, and this deviation between the calculated and experimental pH is worsened as the flow rate decreases."

11.    The average pH along the x-axis at three different sections of the cell of the experiment of acidified pyranine and basified DDW mixing could be added to Figure 5 and discussed in a separate subsection 3.4.

We thank the reviewer for this comment, yet we are not sure it is a good idea to mix figures 5 and 6. Figure 5 is a comparison between the averaging of the pH calculated from the R6G experiments and pH calculated by the acidifed and basified co-flow of the pyranine, while Figure 6 is the measured pH by the co-flow of the acidifed pyranine and basified DDW, making them incompatible. However, we do believe that the comment points to the fact that Figure 5 should be presented in the right context, which is the discussion on the calculated to measured pH, given that Figure 5 is a product of the modified Figure 4. To address this issue of context, we combined the new Figure 4 and Figure 5 into one figure and referred to it as a whole. We also address the combined figure in the text, which provides the right context.

12.    The solutions used in the experiments have different chemical compositions, thus certainly different salinities, viscosities, and densities, affecting flow and mixing. I hope the authors can provide the information and add some comments to the discussion.

We agree with the reviewer that each solution (i.e., R6G, acidify and basified pyranine, basified DDW) will have slight differences in salinities. We address the effect of salinity in the previously referred to as section 3.2.3, where we state the following:

"We repeated this calculation for the hydroxide ions ($OH^-$), the protons ($H^+$), the $Cl^-$ and the Na+, and for all cases, the diffusive flux (marked by the first term in (10) and scales with $D_{OH^-}$ [cm2/s]) was two to three orders of magnitude greater than the electric flux (marked by the second term in (10) and is approximately $J_E (OH^-) \approx 2 \times 10{-7}$ [mol/cm2·s]) due to the ion concentration, making them negligible for our study (see Bard and Faulkner (2001) for details)."

However, the effect of the salinity, and definitely the pH, on density and viscosity is negligible. To reach a noticeable viscosity change (1% of change), one should be at the order of 1g/Kg, where we are at the order of 0.04 and 0.18 for the R6G and pyranine. At these concentrations, their effect on density is negligible. We refer to this in the paper in the following sentence:

"The R6G's concentrations were 2 mg/50 ml double distilled water (DDW with $\approx$ 18 MΩ·cm$^{-1}$ at 25 ∘C, the lab temperature, purified by Milli-Q) for the R6G (corresponding to 0.083 mM) and 9 mg/50 ml DDW for the pyranine (corresponding to 0.347 mM). These concentrations had no measurable effect on the fluid viscosity and density in this experimental setup."

13.    I am surprised by the methodology of calculating a specific Péclet number for each species, rather than calculating effective diffusion coefficients to obtain the Péclet numbers of each solution. Comparing the effective diffusion coefficients would bring more value to the discussion when comparing the results from the experiments of acidified and basified pyranine mixing with acidified pyranine and basified DDW mixing.

The reviewer's comment is well taken, and indeed, the domain porosity should have an influence on the diffusion coefficient. We have done these calculations by scaling the diffusion and Pe with the tortuosity of the domain and adding this analysis to our work. We do find that

there is an obvious scaling with the heterogeneity that is now incorporated in the manuscript, and through this scaling, we have a better estimation of the transverse migration of pH. This scaling with heterogeneity still misses the measured transverse pH migration, but the trend is definitely there. Thank you for proposing this idea. This aspect is addressed in the following sentence in the methods section:

"The Peclet number is a measure of the velocity magnitude ($v_d$), and the diffusion ($D$), which is an intrinsic property of the fluids over the mean pore size (R) (Bossis and Brady (1987)). While the mean pore size remains the same for all heterogeneity, there are small porosity ($\phi$) variations (see Table 1 for details). However, the main heterogeneity effect is on the interface between the co-flowing fluids, forming a torturous path. To address this, we define an effective diffusion coefficient $D_{eff} = \frac{D\phi}{T}$, which scales the diffusion of the reactants in water, as shown in many studies (Ray et al. (2018); Fogler (2011); Guo et al. (2022); Kim et al. (1987); Quintard (1993); Quintard and Whitaker (1993); Beyhaghi and Pillai (2011)). The tortuosity ($T$) can be directly calculated from the normalized standard deviation σ/R, which marks the range for the pillar center movement from a uniform grid using the following relation, $T = 1 + \sigma/R$, and leading to the effective Pe number of:

$$Pe_{eff} = \frac{v_d RT}{D\phi} \tag{2}$$

and scaling the Peclet number as depicted in Table 1.

The fluorescent conservative tracer used for the mixing experiments (Figure 1.a.) is rhodamine 6G (R6G), which is widely used to visualize flow patterns, such as in the domain of environmental hydraulics (Barzan and Hajiesmaeilbaigi (2018)). Pyranine (8-hydroxypyrene-1,3,6-trisulfonate) is used for the reactive and combined experiments (Figure 1.b.-c.) as the pH reactant, as its fluorescent emission spectra and intensity are highly dependent on medium pH (Avnir and Barenholz (2005)), therefore suitable for monitoring pH changes."

We also added a table depicting the results of the scaled Pe and transverse pH migration marked through the mean square displacement (MSD). However, one should be careful since the transverse dispersivity for the R6G is increasing with the heterogeneity, while the Peclet is also increasing, as expected, yet we see that the spread and extent of the diffusion are still the dominant factors for the pH. As such, to account for the heterogeneity as it increases the transverse mixing, Peclet alone is not sufficient to account for pH change. We added the following in section 3.2.2. to address this issue:

"This high diffusion rate leads to a diffusion dominated transverse flux captured by the pH enhanced spread as the applied flux reduces, forming a low Pe over the pore size. Calculating the $OH^-$ transverse migration from the diffusion mean square displacement over the 10 s it takes for the fluid to advance the length of the cell (4.5 mm) for the 10 µL/h flow rate (recall that

the $v_d$ = 0.0142 cm/s), the high diffusion advances the $OH^-$ 0.2 mm. As diffusion is isotropic in nature, it not only occurs transversely to the flow but also aligned with the flow, leading to a steady state of $OH^-$ neutralized by the lower pH and marked by the acidified pyranine, as seen in the homogeneous case (Figure 4c.e). Multiplying this diffusion advancement by the Pe ratio reported in Table 1, brings this diffusion spread to 0.3 mm (see Table 3), nearly covering the full extent of the cell, and similar to the spread in (Figure 4c.e). However, for the same extent of time and Darcy velocity, the high shear in the heterogeneous case further mixes the $OH^-$, leading to full homogenization of the pH in the flow cell (see Figure 4c.h, and Table 3). Yet, the same increase in shear between the homogeneous and the heterogeneous case for the high flux/Pe, produces a smaller relative effect on the $OH^-$ migration (Figures 4c.a-d)."

| $\sigma/R$ $[-]$ | 0.0 | 0.01 | 0.1 | 0.5 |
|---|---|---|---|---|
| $v_d$ $[mm/s]$ | 0.146 | 0.155 | 0.155 | 0.162 |
| $MSD$ $[mm]$ | 0.2 | 0.19 | 0.19 | 0.19 |
| $k \times 10^{-6}$ $[mm^2]$ | 69 | 40 | 34 | 13 |
| $MSD \times Pe_{eff}/Pe$ $[mm]$ | 0.297 | 0.31 | 0.336 | 0.468 |

**Table 3.** The table depicts the Darcy velocity, MSD, and MSD scaled by the effective Peclet ratio for each heterogeneity.

**Technical corrections**

• The abstract is usually a single paragraph.

Agreed and corrected.

• There is a missing space in the text before the left parenthesis of all citations.

Agreed and corrected. We converted the manuscript to Latex, which solved many technical issues.

• Two citations in the same sentence should be grouped in the same parentheses (e.g., line 21).

Agreed and corrected.

• Line 27: the formulation "geological studies, to watershed or global issues" could be clarified.

This sentence was removed to balance the introduction following comment 1 by the reviewer.

• The units are not presented in the SI format (e.g., "µl" should be "µL", "hr" should be "h", "sec" should be "s") and the fraction format should be avoided and replaced with the inline format.

Agreed and corrected.

• The captions of figures 3 to 5 could be shortened since medium heterogeneities are already labeled.

Agreed and corrected.

• Figure 1e: the curves cannot be distinguished based on the chosen colors in the legend, even for non-color-blinded readers.

Agreed and corrected.

• There is a numbering problem for the equation of the ion flux: (8) should be (9).

Agreed and corrected.

• Line 269: misspelled "relay" should be replaced by "rely".

Agreed and corrected.

• Units are missing when presenting the ionic mobility and the electric field (line 300), then for $J_D(OH^-)$ and $J_E(OH^-)$.

Agreed and corrected.

• The guidelines recommend separating the summary from the conclusions.

Done

---

## Author Comment (AC2)

**Reply to reviewer #2**

We thank the reviewer for finding our work interesting and for identifying its relevance to the HESS readership. The comments from the reviewer were seminal in improving our work and mainly in putting it in the right context. These comments also allowed us to implement the simulation results in the main text while commenting on the level of similarity between them and also commented and compared the various electric potentials for various distances in the Nernest-plank equation. We are extremely grateful to the reviewer for the time invested in reviewing our work. We believe that we have responded (marked in blue) to all the comments by the reviewers (marked in black).

The manuscript provides very interesting experiments and interpretations.
However, it requires revision before publication in HESS for the following reasons:
– It should contain a state of the art of the reactive transport experiments
already performed in the past and better highlight the contribution of this
paper (see for example additional paper listed below and references here
in).
We are grateful to the reviewer for providing these references on the Nernest-Planck implementation and validity for flow in porous media. While this is not the main focus of the paper, it is an important aspect in establishing all the contributing factors to the pH migration in our experiments. We have incorporated these references in the introduction and in section 3.2.3.
See lines:
"Specifically for pH reactions, experimental data with high Peclet value for transverse reaction are in good agreement with the Advection-Dispersion-Reaction equation (ADRE), which uses a single diffusion coefficient for all species in a multispecies reactive system \cite{Lawniczak}, especially in stirred flow-cell reactors \cite{Liu2011}",
And
"Simulations performed on experimental results from a Hele-Shaw cell (\cite{Huang 2023, Almarcha 2010}), showed the importance of considering the Coulombic interaction via the Nernest-Plank equation and species specific diffusion coefficient for density-driven flow in bulk. Additional simulations and experiments on porous media, coupling the electrical gradient with various flow configurations, while considering the different diffusion values for each reactive species, yet for high Peclet, pointed to the important role of Coulombic interactions during reactive transport \cite{rolle2018nernst,rolle2013coulombic}, yet it remains to be seen how relevant it is to the system presented in this study."

– The supplementary material should be included in the manuscript. The
processes considered in the modelling should be detailed. Parameters,
boundary conditions, space and time discretizations for the simulation of the
flow and the transport should be given and justified. The differences
between the model and the experimental data should be commented in

details.

Not including the simulation section within the main manuscript is an overlook on our part, and we have moved it to the main manuscript from the supplementary. Moreover, following the reviewer's comment, we added a section to the methods focused on the simulation details (section 2.3). In this section, we describe the simulation type, grid size, equations, and 2D comparison between the model and the results are now stated. The difference between the experimental results and the simulation is now reported in section 3.4. While these simulations do provide the descriptive model for the R6G and pH migration, we are of the opinion that the simulations are not seminal for the main finding in the paper.

"2.3 Comsol simulations

The results for both the mixing and reactive experiments, described in section 2.1, were simulated using the Comsol multi-physics Stokes flow simulator. To that end, the Autocad file with the 2D design and dimensions of the flow cells was imported to the simulator with their dimensions and no slip and no flow boundary condition for the pillars and walls. The inlet and outlet were defined as a Dirichlet boundary condition, corresponding to the constant flux condition imposed by the syringe pump. The simulation followed the following laminar flow equations for an incompressible fluid, namely the continuity, mass conservation, and viscous stress, respectively:

$$\rho\, \partial u/\partial t + \rho(u \cdot \nabla)u = \nabla \cdot [-pI + K] \tag{10a}$$

$$\rho \cdot u = 0 \tag{10b}$$

$$K = \mu(\nabla u + (\nabla u)^T) \tag{10c}$$

Where $\rho$ is the fluid density, $u$ is the velocity in vector form (marked by bold) aligned and transverse ($T$) to the principal flow direction, $\nabla p$ is the pressure drop over the determinant $I$, $K$ is the stress tensor, and $\mu$ is the fluid viscosity. To account for the transport of the R6G and basified solution, the following transport equation is used to account for the concentration ($C_n$) of specific chemical species noted by n:

$$\partial c_n/\partial t + \nabla \cdot J_n + u \cdot \nabla c_n = R_n \tag{11a}$$

$$J_n = -D_n \nabla c_n \tag{11b}$$

Where $J_n$ is the diffusive flux calculated for each chemical species by its corresponding diffusion coefficient, $D_n$, and the chemical retardation factor per species $R_n$. The concentration, $C_n$, is inserted as mol/M3 at the inlets according to the experimental values and as a fixed boundary value.

The maximum and minimum element sizes within the adaptive mesh used for the solid boundaries in the simulation are 1070 and 49.3 µM , while the maximum and minimum element sizes for the fluid calculation are 101 and 4.5 µM , for the adaptive mesh in the finite element linearized calculation. The simulation begins with the introduction of either the R6G or pH difference at the two inlets simultaneously, and allowing the simulation to evolve up to the initial time frame in the experiment stated in section 2.1, namely 5 and 10 minutes for the Darcy velocity of 1.42, and 0.142 mm/s, respectively, while the time discretization ranging between 5 to 15 seconds depending on the level of heterogeneity. The study state flow is achieved extremely fast within the simulation (1 ~ 2 simulated minutes), and therefore, there was no need to run it for another 5 and 10 minutes as in the experiment. These mesh sizes and temporal discretization were optimized to get the best results under the best stability, and

simulations took about 5 minutes on a Core i5, 10 Gen computer with 16 Gig of memory. For each iteration, the concentration, velocity, and pressure were extracted, while the simulated 2D spatial distribution for the R6G and pH were compared with the experimental values using the 2D R2 function in Matlab."

– The concentration normalization assumes a linear relationship between intensity and concentration (eq. 5). Please comment.

We thank the reviewer for providing us the opportunity to clarify this point. As stated in section 2.2.1, the normalization of the R6G concentration is based on the Beer-Lambert law that linearly relates the intensity with the concentration. As stated at the end of that paragraph, this relation was also verified in other studies but also experimentally verified for our setup by saturating the flow cell with DDW with a known concentration quantity of R6G and imaging it through the microscope.

– What is the reliability of an experiment, i.e., what is the difference in the transverse missing/dispersion when repeating a reactive transport experiment?
The reviewer is right, and the reliability analysis reported in response to reviewer 1 comment did not separate between the R6G and the pyranine pH measurements. In fact, the same analysis is done for the R6G, and for the pH experiments, recall that the intensity value is used to infer both R6G concentration and pH value. That is why we reported it in 2.2 and not in the specific subsection for mixing and reactive experiments. We now corrected this issue by clarifying in the text that this test was done for both experiments:
"As such, this intensity analysis, which provides both the error bounds and repeatability of the layout, is done for both the R6G and pH experiment."

– I do not understand why the distance between the two inlets is used for the computation of the ion fluxes due to non-neutral electric field generated by differences in the diffusion coefficients (Nernst-Planck equation). This gradient should be a local one (not at the scale of the setup), that can be computed using the images. Please clarify and comment.
This is an interesting point raised by the reviewer, and it refers to the calculation of both the diffusion component in the Nenest-Plank equation and the Coulombic interaction, as shown in Rolle et al 2017. As such, the $\Delta x$ term is used for both the concentration difference for the diffusion and for the electric field derived from the electrostatic potential. Therefore, changing this value for one term will similarly change the second term, leaving the ratio between them the same. As such, we are not sure this will change the overall effect. Then again, it could very well be that we misunderstood the reviewer's comment, and if so, we humbly ask for some more clarification.

Minor comments:
What is the pixel size of the images?

The magnification used for these experiments is 2X (now stated in section 2.2), and therefore each pixel is a 3.3 $\times$ 3.3 $\mu M$ per pixel. We added this to the manuscript:
"All experimental images taken by the confocal are taken by a Prime BSI camera with a 95 % quantum efficiency and $1e^-$ median noise, with an exposure time of 500 ms, bit depth of 16-bit, and magnification of 2x, providing a resolution of 3.3 $\times$ 3.3 $\mu M$ per pixel"

What is the reliability of the concentration values (0.1 % camera noise but what about the uncertainty related to the calibration)?
The uncertainty of the calibration is done by the same uncertainty measurement done for the image analysis stated in section 2.2:
"For the 100 µL/h flow rate, a series of 50 pictures were taken 5 minutes after forming a stable interface between the fluids. Then, after an additional 5 minutes of delay, another series of 50 pictures is taken, under the same conditions. The two series of images are compared to verify the stability of the interface. For the 10 µL/h flow rate, the same imaging sequence was performed, with an initial time of 10 minutes and a subsequent delay time of 10 minutes. For both flow rates, the variance of each pixel intensity (marked as $I_{ij}$, for location ij) for the 50 picture sequences taken for each time, did not exceed 1%. To verify that the interface among image sequences for different times is stable, the criteria were set that the difference between the initial and later imaging sequence that exceeded the 0.1% (white noise of the camera) was averaged in absolute terms, and the stability of the interface was established if the average difference was isotropic and smaller than 1% (namely, , $\langle \frac{|I_{ij}(t=5) - I_{ij}(t=10)|}{|I_{ij}(t=10)|} > 0.1\% \rangle < 1\%$) a similar analysis was performed around the interface to verify that the 1% difference is not the outcome of the bulk behavior."

Please check units through the document and use SI format.
Done
L108: "simultaneous" -> "simultaneous"
Done
L269: "relay" -> "rely"
Done
L133: a character space is missing "solution(Barzan and Hajiesmaeilbaigi, 2018)".
Done
Same for L140, L194.
Done
Two "=" in equation (8) which should be (9).
Done

References
Huang et al., 2023, doi.org/10.5194/gmd-16-4767-2023
Loyaux-Lawniczak S. et al. (2012) DOI: 10.1016/j.jconhyd.2012.06.005
Rolle et al., 2013. doi.org/10.1016/j.gca.2013.06.031
Liu, C. et al., 2011, doi.org/10.1029/2011WR010575
Rolle et al.,2018, doi.org/10.1002/2017WR022344